# How does the OH reactivity affect the ozone production efficiency:case studies in Beijing and Heshan, China

Yudong Yang[1], Min Shao[1, *], Stephan. Keβel[2], Yue Li[1], Keding Lu[1], Sihua Lu[1], Jonathan Williams[2], Yuanhang Zhang[1], Liming Zeng[1], Anke C. Nölscher[2, #], Yusheng Wu[1], Xuemei Wang[3], Junyu Zheng[4],

[1] State Joint Key Laboratory of Environmental Simulation and Pollution Control, College of Environmental Science and Engineering, Peking University, Beijing, China
[2] Department of Atmospheric Chemistry, Max Plank-Institute for Chemistry, Mainz, Germany
[3] School of Atmospheric Science, Sun Yat-Sen University, Guangzhou, China
[4] School of Environmental Science and Engineering, South China University of Technology, China
# now at: Division of Geological and Planetary Sciences, California Institute of Technology, Pasadena, USA

* Corresponding to: Min Shao
Email address: mshao@pku.edu.cn

## Abstract

Total OH reactivity measurements were conducted on the Peking University campus, Beijing in August 2013 and in Heshan, Guangdong Province from October to November 2014. The daily median OH reactivity were $20 \pm 11$ s$^{-1}$ in Beijing and $31 \pm 20$ s$^{-1}$ in Heshan respectively. The data in Beijing showed a distinct diurnal pattern with the maxima over 27 s$^{-1}$ in early morning and minima below 16 s$^{-1}$ in the afternoon. The diurnal pattern in Heshan was not as evident as in Beijing. Missing reactivity, defined as the difference between measured and calculated OH reactivity, was observed at both sites, with 21% missing in Beijing and 32% missing in Heshan. Unmeasured primary species, such as branched-alkenes could contribute to missing reactivity in Beijing, especially in morning rush hours. An observation-based model with the RACM-2 (Regional Atmospheric Chemical Mechanism version 2) was used to understand the daytime missing reactivity in Beijing by adding unmeasured oxygenated volatile organic compounds and simulated intermediates of the degradation from primary VOCs. However, the model could not find the convincing

explanation for the missing in Heshan, where the ambient air was found to be more aged, and the missing reactivity was presumably attributed to oxidized species, such as unmeasured aldehydes, acids and di-carbonyls. The ozone production efficiency was 21% higher in Beijing and 30% higher in Heshan when the model was constrained by the measured reactivity, compared to the calculations with measured and modeled species included, indicating the importance of quantifying the OH reactivity for better understanding ozone chemistry.

## 1. Introduction

Studies on total OH reactivity in the atmosphere have been of increasing interest over the last two decades. The instantaneous total OH reactivity, is defined as

$$k_{OH} = \sum_i k_{OH+X_i} [X_i] \tag{1-1}$$

where X represents a reactive species (CO, NO$_2$ etc.) and $k_{OH+X_i}$ is the rate coefficient for the reaction between X and OH radicals. Total OH reactivity is an index for evaluating the amounts of reductive pollutants in terms of ambient OH loss and hence their roles in atmospheric oxidation (Williams, 2008; Williams and Brune, 2015; Yang et al., 2016). It also provides a constraint for OH budget calculation in both field campaigns and laboratory studies (Stone et al., 2012; Fuchs et al., 2013).

Total OH reactivity measuring techniques, e.g., two laser-induced-fluorescence (LIF) based techniques (Calpini, et al., 1999; Kovacs and Brune, 2001) and one proton-transfer-reaction mass spectrometry (PTR-MS) based technique, comparative reactivity method (CRM) (Sinha et al., 2008) were developed in recent years. A brief comparison of these techniques and their interferences were summarized (Yang et al., 2016). By deploying these measuring techniques, total OH reactivity measurements have been intensively conducted in urban and suburban areas. Details of these campaigns were listed in Table 1 and Table 2. Most of the campaigns exhibited similar diel features with higher reactivity in dawn and rush hours of early morning, and lower levels in the afternoon, which could be explained by the change in

boundary layer height, emissions and oxidation processes. Anthropogenic volatile
organic compounds (VOCs) and inorganics, such as CO and $NO_x$ (NO + $NO_2$) are
major known OH sinks in urban areas.
However, a substantial difference between measured and calculated or modelled
OH reactivity, termed as the missing reactivity, was revealed in most field campaigns.
Compared to the high percentages of missing reactivity in forested areas (Sinha et al.,
2010; Nölscher et al., 2012; 2016; Edwards et al., 2013, Williams et al., 2016), most
campaigns in urban and suburban areas gave relatively lower percentages of missing
reactivity except for the 75% missing reactivity in Paris in MEGAPOLI under the
influences of continental air masses (Dolgorouky et al, 2012).
Various methods were used in exploring the origins of missing reactivity.
Unmeasured primary species are important candidates. Sheehy et al. (2010)
discovered a higher percentage of missing reactivity in morning rush hours and found
that the unmeasured primary species, including organics with semi and low-volatility,
could contribute up to 10% of total reactivity. Direct measurements on reactivity of
anthropogenic emission sources were conducted, such as vehicle exhaust and gasoline
evaporation. An average of 17.5% missing reactivity was found in vehicle exhaust
measurements (Nakashima et al., 2010). For gasoline evaporation, a study showed
that if primary emitted branched-chained alkenes were considered, the measured and
calculated reactivity then agreed (Wu et al., 2015). Besides primary emitted species,
unknown secondary species were not negligible. Yoshino et al. (2006) found a good
correlation between missing reactivity and measured oxygenated VOCs (OVOCs) in
three seasons except for winter, assuming that the unmeasured OVOCs could be
major contributors of missing reactivity, in one case the OVOCs could increase
reactivity by over 50% (Lou et al., 2010). The observation-based model (OBM) was
widely used to evaluate the measured reactivity (Lee et al., 2010; Lou et al., 2010;
Whalley et al., 2016), confirming the important contribution from OVOCs and
undetected intermediate compounds,.
Ground-level ozone pollution has been of increasing concerns in China. While
the ozone concentration exceeds Grade II of China National Ambient Air Quality

Standards (2012) (93 ppbV) frequently in summer in Beijing-Tianjin-Hebei area and Pearl River Delta (PRD) region (Wang et al., 2006; Zhang et al., 2008), it appears there is an increasing trend for ozone in Beijing and other area recent years (Zhao et al., 2009; Zhang et al., 2014). Comparing to traditional empirical kinetic model approach (EKMA) (Dodge et al., 1977), the OH reactivity due to VOCs (termed as VOCs reactivity) rather than VOCs mixing ratio was used in the calculation of ozone production rate (Geddes et al., 2009; LaFranchi et al., 2011; Sinha et al, 2012; Zhang et al., 2014). Due to the limitation of current measurement techniques, some VOCs species which could not be quantified so far, and therefore cannot be integrated into current chemical mechanisms of model run, could laid a great uncertainty in ozone production prediction. By directly measuring the total OH reactivity, VOCs reactivity can be obtained by deducting the inorganic reactivity from the total OH reactivity, which provides a constrain for evaluating the roles of reactive VOCs in air chemistry (Sadanaga et al, 2005; Sinha et al, 2012; Yang et al., 2016).

This paper presents field data in China from two intensive observation conducted in August 2013 in Beijing, and October to November 2014 in Heshan, Guangdong, focusing on OH reactivity and related species. The variations of total OH reactivity at both sites were compared with similar observations in urban and suburban areas worldwide. Thereafter, a zero dimensional box model based on Regional Atmospheric Chemical Mechanism 2 (RACM2) was employed for OH reactivity simulations. The possible missing reactivity and its importance for the ozone production calculation are discussed.

## 2. Methods

### 2.1 Total OH reactivity measurements
### 2.1.1 Measurement principles

Total OH reactivity was measured by the comparative reactivity method (CRM) first developed at Max Planck Institute for Chemistry (Sinha et al., 2008). The CRM system was built accordingly in Peking University, which consisted of 3 major components: inlet and calibration system, reactor, and measuring system as shown in

Fig 1. Ambient air was sampled after a teflon filter and then pumped through a 14.9m
Teflon 3/8 inch (outer diameter) inlet at about 7 L·min$^{-1}$ rate, with a 5 - 6 s residence
time.
In this method, pyrrole ($C_4H_5N$) was used as the reference substance and was
quantified by a quadrupole PTR-MS (Ionicon Analytic, Austria). There are 4 working
modes for measuring procedure: In the C0 mode, pyrrole (Air Liquid Ltd, U.S.) is
introduced into the reactor with dry synthetic air (99.99%, Chengweixin Gas Ltd,
China). A mercury lamp (185nm, used for OH radicals generation) is turned off and
high-pure dry nitrogen (99.99%, Chengweixin Gas Ltd, China), is mixed into the
reactor through a second arm. In this mode, the highest signals of m/z 68 (protonated
mass of pyrrole) c0 are obtained. Then in the C1 mode, the nitrogen and synthetic air
is still dry but the mercury lamp is turned on. The mixing ratio of pyrrole decreased to
c1. The difference between c0 and c1 is mainly due to the photolysis of pyrrole (Sinha
et al., 2008). C2 mode is the "zero air" mode in which synthetic air and nitrogen are
humidified before being introduced into the reactor. The photolysis of water vapor
generates OH radicals which react with pyrrole in the reactor to c2 level. Then C3
mode is the measuring mode in which the automatic valve switches from synthetic air
to ambient air. The ambient air is pumped into the reactor to react with OH radicals,
competing with pyrrole molecules. The mixing ratio of pyrrole is detected as c3. Total
OH reactivity is calculated as below, based on equations from Sinha et al. (2008):
$$k_{OH} = c1 \times k_{Pyr+OH} \times \frac{c3-c2}{c1-c3} \qquad (2\text{-}1)$$

Ambient air or synthetic air was introduced at 160 -170 ml min$^{-1}$ with the total
flow 320 – 350 ml min$^{-1}$(The typical dilution factor was about 2-2.15 depending on
the situation). The residence time of air inside the reactor was less than 30 s before
they were pumped by the Teflon pump. The typical c1 mixing ratio for pyrrole in
Beijing and Heshan measurements were about 60 ppbV and 55 ppbV, while the
mixing ratios of OH radicals generated by mercury lamp were about 35 ppbV and 28
ppbV. The mixing ratios were quite consistent for either of the campaigns,
respectively. Corrections about pseudo-first order kinetics were conducted for both
measurements, based on the methods in Sinha et al (2008). The typical correction
factors could be presented as

$$R_{true} = 0.0008 * (R_{mea})^2 + 0.78 * R_{mea} - 0.042 \qquad (2\text{-}2)$$

$$R_{true} = -0.0004 * (R_{mea})^2 + 0.81 * R_{mea} - 0.017 \qquad (2\text{-}3)$$

**2.1.2 Calibrations and tests**
We performed two calibrations for the measurements. First, PTR-MS was
calibrated by diluted dry pyrrole standard gas ranging from less than 10 ppbV to over
160 ppbV (presented in Fig S1). Additionally, we conducted an inter-comparison with
humidified pyrrole dilution gas. The sensitivity was about 3% to 5% higher than dry
calibration, which was considered for later calculation (Sinha et al., 2009). The tests
of the CRM system were done by using both the single standard gas, such as CO,
propane, propene (Huayuan Gas Ltd, China) and a standard of the mixture of 56
non-methane hydrocarbons (NMHCs) (SpecialGas Ltd, U.S.). The results of the
calibrations and tests were presented in Fig 2. Measured and calculated OH reactivity
agreed well within the uncertainty for all calibrations.
A key factor influencing the measurement results is the stability of OH radical
generator. One major interference could be the difference in relative humidity
between C2 mode and C3 mode. During the experiment, we used one single needle
valve to control the flow rate of synthetic air going through the bubbler, so that the
relative humidity during C2 mode could be adjusted to match humidity during
ambient sampling (C3 mode). Meanwhile, the remaining minor difference could be
corrected by factors derived from the OH reactivity-humidity correction experiment.
The details of the OH-correction experiment and the data were presented in the
supporting information (Fig. S1 and S2).
The other interference might be caused by ambient NO, which produces
additional OH radicals via recycling of $HO_2$ radicals (Sinha et al., 2008; Dolgorouky
et al., 2012; Michoud et al., 2015). The amount of OH radical through this pathway is
hard to be quantified. In the morning rush hours or on polluted cloudy days, NO
levels could rise to over 30 ppbV in both Beijing and Heshan, which could then
potentially introduce high uncertainties for measurements. The NO-correction

experiments were conducted by introducing given amounts of VOCs standard gases into the reactor. When the stable concentrations for c2 were reached, different levels of NO were injected into the reactor and the "measured" reactivity decreased as the NO mixing ratio increased. Then a correction curve was fitted between the differences in reactivity and NO mixing ratios. Several standard gases and different levels of base reactivity (from less than $30s^{-1}$ to over $180s^{-1}$) have been tried and the curve was quite consistent for all tested gases, as shown in Fig 3. The correction derived from the curve was used later to correct ambient measurements according to simultaneous detected NO levels. The correction was necessary when NO mixing ratio was larger than 5 ppbV, which was quite often observed in the morning time as well as cloudy days in Beijing and Heshan. The relative change for reactivity results could be over $100 \ s^{-1}$ when NO mixing ratio was about 30 ppbV.

A further potential interference comes from nitrous acid (HONO). The photolysis of HONO in the reactor could generate the same amount of OH radicals and NO molecules, as shown in R1. The additional OH radicals and NO molecules can be both interferences with the reactivity measurements. Similar correction experiments were conducted as the NO correction experiment. HONO were added stepwise in several mixing ratios (1-10 ppbV), generated by a HONO generator (Liu et al., 2016) and thus introduced into the reactor. A curve was fitted between the differences in reactivity and HONO mixing ratios, as presented in Fig 4. The correction associated with this curve was also applied later in the ambient measurements.

$$HONO \xrightarrow{h\gamma} OH+NO \qquad (R1)$$

To make sure the production of OH radicals was stable during the experiments, C1 mode was measured for 1-2 hour every other day and C2 mode was measured for 20-30 minutes every two hours. With above calibrations and tests into consideration, the detection limits of CRM methods in two campaigns was around $5 \ s^{-1}$ ($2 \sigma$). The total uncertainty of the method was about 20% ($1 \sigma$), due to rate coefficient of pyrrole reactions (15%), flow fluctuation (3%), instrument precision (6% when measured reactivity $> 15 \ s^{-1}$), standard gases (5%) and corrections for relative humidity (5%).

**2.2 Field measurements**

**2.2.1 Measuring sites and periods**

The urban measurements started from August $10^{th}$ to August $27^{th}$, 2013 at Peking University (PKU) Site (116.18°E, 39.99°N), which was set on the roof of a 6-floor building. The site is about 300 m from the 6-lanes road to the east and 500 m to the 8 -lanes road to the south. This site is an urban site used for intensive field measurements of air quality in Beijing for long. Detailed information about this site can be found elsewhere (Yuan et al., 2012).

Suburban measurements were conducted from October $20^{th}$ to November $22^{nd}$ 2014 at Heshan (HS) site, Guangdong (112.93°E, 22.73°N). The site is located on top of a small hill (60 m above ground) in Jiangmen, which is 50km from a medium size city Foshan (with a population of about 7 million) and 80 km from a megacity Guangzhou. This is the super-site for measurements of air quality trends by Guangdong provincial government, detailed information about which can also be found in Fang et al (2016).

**2.2.2 Simultaneous measurements**

During both intensive campaigns, fundamental meteorological parameters and trace gases were measured simultaneously. Meteorological parameters, such as temperature, relative humidity, pressure, wind speed, wind direction were measured. NO and $NO_x$ mixing ratios were measured by chemi-luminescence (model 42i, Thermo Fischer Inc, U.S.), and $O_3$ was measured by UV absorption (model 49i, Thermo Fischer Inc, U.S.). CO was measured by Gas Filter Correlation (model 48i, Thermo Fischer Inc, U.S.), and $SO_2$ was measured by pulsed fluorescence (model 43C, Thermo Fischer Inc, U.S.). The photolysis frequencies were measured by a spectral radiometer (SR) including 8 photolysis parameters. These parameters were all averaged into 1-minute resolution. The performances of these instruments were presented in Table S1 and Table S2.

VOCs were measured by a cryogen-free online GC-MSD/FID system, developed by Peking University (Yuan et al., 2012; Wang et al., 2014a). The time resolution is 1

hour but the sampling time starts from the 5th minute to 10th minute every hour. The
system was calibrated by two sets of standard gases: 56 NMHCs including 28 alkanes,
13 alkenes and alkynes, 15 aromatics; EPA TO-15 standards
(http://www.epa.gov/ttnamti1/les/ambient/airtox/to-15r.pdf), including additional
OVOCs and halocarbons. The detection limits ranged from 10ppt-50ppt, depending
on the species. Formaldehyde was measured by the Hantzsch method with time
resolution of 1 minute. Detailed information about this instrument is described in one
previous paper (Li et al., 2014).
**2.3 Model description**
**2.3.1 Box model**
A zero-dimensional box model was applied to produce the unmeasured secondary
products and OH reactivity for both field observations. The chemical mechanism
employed in the model was RACM2 (Stockwell et al., 1997, Goliff et al., 2013), with
implementation of Mainz Isoprene Mechanism (MIM, Pöschl et al., 2000) and update
versions by Geiger et al. (2003) and Karl et al. (2006) for isoprene reactions. The
model was constrained by measured photolysis frequencies, ancillary meteorology
and inorganic gases measurements, as well as VOCs data. Mixing ratios of methane
and $H_2$ were set to be 1.8 ppmV and 550 ppbV. The model was calculated in a
time-dependent mode with 5-min time resolution. Each model run started with 3 days
spin-up time to reach steady-state conditions for long-lived species. Different
scenarios with 1 day, 2 days and 3 days spin-up time have been tried while the
differences were within 10%. Additional loss by dry deposition was assumed to have
a corresponding lifetime of 24 hours to avoid the accumulation of secondary
productions. The boundary layer height was set as constant as 1000 m in the model
due to the lack in measurements. This was similar to model setups in Lu et al (2013)
and field measurement results in Guo et al (2016).
**2.3.2 Ozone production efficiency**
Ozone production efficiency (OPE) is defined as the number of molecules of
total oxidants produced photochemically when a molecule of $NO_x$ was oxidized
(Kleinman, 2002, Chou et al., 2011). It helps to evaluate the impacts of VOCs
reactivity on ozone production in various $NO_x$ regimes. In this work, the OPE was
expressed as the ratio of ozone production rate (i.e. $P(O_3)$) to $NO_x$ consumption rate
(i.e. $D(NO_x)$). $NO_z$, calculated as the difference between $NO_y$ (sum of all odd-nitrogen
compounds) and $NO_x$, was assumed to be the oxidation products of NOx. Thus the
OPE could be also calculated as $P(O_3)/P(NO_z)$. The ozone production rate is obtained
as 2-2, and the $P(NO_z)$ is approximately as production rate of $HNO_3$ as well as the
production rate of organic nitrate, which is given as 2-3.
$$P(O_3) = k_{HO_2+NO}\,[HO_2][NO] + \sum_i k_{RO_{2_i}+NO}\left[RO_{2_i}\right][NO] \qquad (2\text{-}2)$$

$$P(NO_z) = k_{NO_2+OH}\,[NO_2][OH] + \sum_i k_{RO_i+NO_2}\,[RO_i][NO_2] \qquad (2\text{-}3)$$

## 3. Results

### 3.1 Time series of meteorology and trace gases

The time series of selected meteorological parameters and inorganic trace gases
were presented in 5 minute averages (Fig 5). The median values of the inorganic trace
gases were $0.715 \pm 0.335$ ppmV for CO, $6.3 \pm 5.75$ ppbV for NO and $36.5 \pm 21.3$
ppbV for $NO_2$, $57 \pm 44$ ppbV for $O_3$ in Beijing. In Heshan, the median results were
$0.635 \pm 0.355$ ppmV for CO, $9.7 \pm 6.95$ ppbV for NO, $29.6 \pm 12.6$ ppbV for $NO_2$, and
$55.7 \pm 34.9$ ppbV for $O_3$. Both results were within the range of data from literatures
(Zhang et al., 2008; Zheng et al., 2010; Zhang et al., 2014). However, daytime
averaged $O_3$ mixing ratio in Beijing 2013 was a little lower than the medium results
(about 60 ppbV) in normal years (Zhang et al., 2014). This could be due to higher
frequencies of cloudy and rainy days, which accounted for about 1/3 of our
measurement duration. The measured maximum photolysis rates in cloudy/rainy days
were about half of peak values of $J(O^1D)$ on sunny days. Even under this
circumstances, ozone levels from the campaign remained high, the pollution episodes
with ozone exceeding Grade II of China National Ambient Air Quality Standards (93
ppbV) occurred quite often, and the percentage of exceedance were 40% in Beijing
and 20% in Heshan.
The mixing ratios of VOCs in both campaigns were presented in Table S3 and
Table S4. In summer Beijing, alkanes accounted for over 60% of the summed VOCs
mixing ratios during most of the time, while in Heshan the contribution from
aromatics was 6% higher than that in Beijing. This could be explained by stronger
emissions from solvent use and paint industry in the PRD region (Zheng et al., 2009).
The ratio of toluene to benzene, which is typically used qualitatively as an indicator
for aromatics emission sources also supported this assumption. While this ratio in
Beijing was close to 2, similar to vehicle emissions (Barletta et al., 2005), the ratio in
Heshan was higher than 3 due to strict control of benzene in solvent usage these days
(Barletta et al., 2005; Liu et al., 2008). In the ozone polluted episode in Fig 5, the
mixing ratios of most species were about twice to three times higher than the daily
average results.

The diurnal variations of $NO_x$, $O_3$ and photochemical age from Beijing and

Heshan site were compared in Fig 6 and Fig 7. Both sites presented similar diurnal
patterns for $O_3$ and NO. However, the highest 1-hour average $O_3$ value at PKU site
came in the afternoon and stayed at high level till the dawn. While $O_3$ pattern at
Heshan site did not stay high in the afternoon. An additional similarity was that the
NO peaks occurred at similar times for both sites. But NO decreased at a slower rate
in Heshan till even 12:00 p.m. This was likely explained by the facts that the NO
observed at PKU site was mainly from local vehicle emissions while $NO_x$ at Heshan
site was significantly influenced by long-range transported of air masses.

VOCs measurements provided us chance to evaluate the oxidation state at two

sites. Based on the OH exposure calculation methods (de Gouw et al., 2005), we
chose a pair of VOCs species: m,p-xylene and ethylbenzene to calculate the
photochemical age:
$$[OH]\Delta t = [\ln(\frac{[E]}{[X]})_t - \ln(\frac{[E]}{[X]})_0]/(k_E - k_x) \tag{3-1}$$

Here, [E] and [X] represents the mixing ratios of ethylbenzene and m,p-xylene,

$k_E$ and $k_X$ means the OH reaction rate coefficient of ethylbenzene and m,p-xylene. As
presented in Fig 7, we chose 1.15 ppbV ppbV$^{-1}$ and 2.3 ppbV ppbV$^{-1}$ as emission
ratios of ethylbenzene to m,p-xylene in Beijing and Heshan, as they were the largest
ratios in diurnal variations for the campaign. The largest OH exposure in Beijing 2013
was calculated as $0.71 \times 10^{11}$ molecule s $cm^{-3}$ in 13:00 LTC, while the largest OH
exposure in Heshan 2014 was calculated to be $1.69 \times 10^{11}$ molecule s $cm^{-3}$ in 14:00
LTC. The results in Beijing were comparable to previous reports (Yuan et al., 2012).
Assuming the daytime average ambient OH concentration was $5.2 \times 10^6$ molecule
$cm^{-3}$ (Lu et al., 2013), the photochemical age in Beijing was estimated to be not more
than 3.5 h. With measured daytime average OH concentration as $7.5 \times 10^6$ molecule
$cm^{-3}$ in Heshan (Tan et al., in preparation), the photochemical age in Heshan was
about 6 h to 7 h, which was about twice the photochemical age of the Beijing
observations, indicating a more aged atmospheric environment in Heshan. However,
the assumed OH radical concentrations' influence on the photochemical age results
should not be neglected.
**3.2 Measured reactivity**

Total OH reactivity ranged from less than 10 $s^{-1}$ to over 100 $s^{-1}$ in Beijing (Fig

5a). The daily median value was $20 \pm 11$ $s^{-1}$. The diurnal patterns changed
significantly from day to day (Fig 8). The averaged diurnal pattern showed that the
total OH reactivity was higher from dawn to morning rush hours with a peak hourly
mean of 27 $s^{-1}$, and decreased to a lower value, median value of 17 $s^{-1}$ in the afternoon.
This diurnal pattern was similar to the variations of $NO_x$ mixing ratios (Williams et al.,

2016).

Meanwhile, measured total OH reactivity in Heshan was higher in median but

the diel variation was less evident. The daily median value was $31 \pm 20$ $s^{-1}$. The OH
reactivity was much less variable in the daily variation. This could possibly due to the
more aged air masses in Heshan, as presented in 3.1. The other probable explanation
could be the two periods of clean air we encountered, during which ground-level
ozone and $PM_{2.5}$ concentrations were rather low, each of the cases lasted for about 5
days during our measurements. And 2 pollution episodes were identified between
October 24th to 27th and November 14th to 17th, 2014. Both episodes showed
accumulation of ozone and $PM_{2.5}$. The total OH reactivity level also built up
significantly (Fig 5b).
**3.3 Variations in missing reactivity**
Significant differences were found between the measured reactivity and
calculated reactivity which derived from mixing ratios of different species multiplied
by their rate coefficients with OH radicals, as presented in Fig 5a for Beijing and Fig
5b for Heshan. Taking all measured species into consideration, $NO_x$ and NMHCs
showed the largest contribution, 45%-55% of total OH reactivity (Fig 9). The OVOCs
had also significant contribution, and measured OVOCs had a sharing of 10% in total
reactivity in Beijing while 7% in Heshan.
The Missing reactivity was on average over 4 $s^{-1}$, 21 ± 17 % of the total OH
reactivity in Beijing and 10 $s^{-1}$, 32 ± 21% in Heshan. The missing reactivity presented
different temporal patterns. In Beijing, the missing reactivities were high during
pollution episodes, especially in the morning rush hours. The percentage of missing
reactivity could reach over 50%. For the Heshan site, the missing reactivity was more
or less stable during the entire campaign. Even in clean days with reactivity levels
lower than 20 $s^{-1}$, 20%-30% of missing reactivity still existed.

## 4. Discussion

### 4.1 Reactivity levels in Beijing and Heshan

The measured VOCs reactivity (obtained by subtracting inorganic reactivity from
total OH reactivity), 11.2 $s^{-1}$ in Beijing and 18.3 $s^{-1}$ in Heshan (Fig 10), was actually
not at high end comparing with the levels from literatures. Tokyo presented a similar
level of VOCs reactivity (Yoshino et al., 2006). The measured NMHCs levels
(obtained by adding all hydrocarbon mixing ratios together) were also not very high,
with Beijing 2013 being around 20 ppbV and Heshan 2014 higher than 35 ppbV. The
relative VOCs reactivity, defined by the ratio of the VOCs reactivity to the measured
NMHCs levels, the values for both Beijing and Heshan were very high.
One possible explanation is the higher content of reactive hydrocarbons in China.
Compared to other campaigns, both sites had higher loading of alkenes and aromatics
(Yuan et al., 2012; Wang et al., 2014b). The other probable reason is the contribution
from OVOCs. In Beijing and Heshan, ambient formaldehyde could accumulate to
over 10 ppbV, which was significantly higher than levels found in other observations
(Li et al., 2013; Chen et al., 2014). Another possible explanation is unmeasured
species, either primary hydrocarbons or secondary products, which will be discussed
in later sessions.
**4.2 Contributions to the missing reactivity: primary VOCs**
As missing reactivity was observed at Beijing and Heshan site, the species
possibly causing these missing were examined. Throughout the whole campaign at the
PKU site, missing reactivity was normally found in the morning, as for an example in
August 16th and 17th 2013 in (Fig 11). Between 5 a.m. to 10 a.m., local vehicle-related
sources were strong, and the chemical reactions were not active yet, and the oxidants
levels thus the secondary VOC species remained low. We assumed that the
unmeasured primary VOCs species could most likely be the major contributors to
missing reactivity. Special attention was paid to the unmeasured branched-alkenes for
their high reactivity and was previously observed from vehicle exhaust (Nakashima et
al., 2010) and gasoline evaporation emissions (Wu et al., 2015). We found only one
dataset of branched alkenes measurements in Beijing, 2005 measured by NOAA (Liu
et al., 2009). We chose the diurnal patterns of missing reactivity in Beijing in 2013
and compared to the diel cycles of four measured branched-alkenes in 2005. The
correlations were found as presented in Fig 11. Considering the contribution of the 4
branched alkenes, the VOCs reactivity could be enhanced by 2.3 s$^{-1}$. This could only
partially explain the missing VOCs reactivity which was around 10 s$^{-1}$. With observed
decreasing trends in mixing ratios of most NMHCs species in Beijing (Zhang et al.,
2014; Wang et al., 2015), the branched-alkenes were insufficient to tell the full story
of the missing reactivity. Unmeasured semi-volatile organic compounds (SVOCs) and
intermediate volatile organic compounds (IVOCs), such as alkanes between C12 to
C30, and polycyclic aromatic hydrocarbons (PAHs) could be also important. Sheehy
(2010) found SVOCs and IVOCs contributed to about 10% in morning time in
Mexico City. Future studies with a wider range of reactive VOCs measurement for
total OH reactivity closure is needed.
**4.3 Contributions to the missing reactivity: secondary VOCs**
Due to limitations in chemistry mechanisms as well as measuring techniques,

secondary products are not fully quantified in ambient air and could probably contribute significantly to the observed missing reactivity, especially in the urban or suburban sites receiving chemically complex aged air masses.

Besides the large missing reactivity during the morning rush hour, there was about 25% difference between measured and calculated reactivity from August 16[th] to 18[th], 2013 at PKU site. Considering high levels of oxidants in daytime, the mixing ratios of branched-alkenes could be lower than 0.1 ppbV, which could not explain the observed missing reactivity. A box model was deployed to investigate the role of secondary species in variation of VOCs reactivity. The model, constrained by measured parameters (meteorology, inorganic gases, VOCs including measured carbonyls), gave the results of VOCs reactivity which agreed well with the measured reactivity in most of the daytime (Fig 11). Major contributors from modeled species were unmeasured aldehydes, glyoxal and methyl glyoxal. Average values of major secondary contributors to modelled reactivity were provided in Table S5. However, the missing in morning hours remain unsolved: In the model run, the higher secondary contribution on August 17[th] 2013 morning was owing to isoprene oxidation products, by using 1.5 ppbv of isoprene levels as model input, the missing reactivity kept over 40% around 7:00 and 8:00 a. m.

The similar model was applied for the Heshan observation (Fig 12). During the polluted episode between October 24[th] and 27[th] 2014, a 30% missing reactivity existed for most time. Unfortunately, the modeled reactivity was only 10-20% higher than calculated reactivity, and not enough to explain the measured reactivity. The major contributors among modeled species were also unmeasured aldehydes, glyoxal, methyl glyoxal and other secondary products, as shown in Table S6. Due to strong emissions of aromatics from solvent use and petroleum industry in PRD region (Zheng et al., 2009), high levels of glyoxal and methyl glyoxal in this region were observed from satellite measurement (Liu et al., 2012) and ground measurements (Li et al., 2013). Compared to the 2006 measurements in Back garden, a semi-rural site in PRD region, the modeled glyoxal was twice as high as around 0.8 ppbV (Li et al., 2013). This difference possibly resulted from higher levels of precursors in 2014

measurements, where the measured reactivity was about 50% higher than the results
in Backgarden 2006 (Lou et al., 2010).
**4.4 Implications for ozone production efficiency**

The investigating of missing in VOCs reactivity is expected to better understand

the ozone formation processes. To evaluate this contribution, we employed the box
model to calculate the influence of VOCs reactivity on OPE. We set two scenarios for
the model run: 1) The base run was constrained with measured species, including all
inorganic compounds, PAMS 56 hydrocarbons, TO-15 OVOCs and formaldehyde.
This is how we obtained the modelled reactivity as presented above, and the
intermediates and oxidation products were reproduced as well. 2) The other scenario
used measured reactivity as a constraint. Due to the difference between measured and
modeled reactivity, we allocated the missing reactivity into several groups. For the
primary species, we assumed the ratio between total chain-alkenes and
branched-alkenes were the same in Beijing 2013 and in Heshan 2014 as the ratio in
Beijing 2005, so we got the assumed mixing ratios of branched-alkenes at both sites.
For secondary species, we allocated the remaining missing reactivity into different
intermediates or products based on weights obtained in the model base run. Under
both assumptions, we ran the OBM and calculated the OPE, as presented in Fig 13.

For both sites, the OPE constrained by measured reactivity were significantly

higher than the OPE we calculated from modeled reactivity. In Beijing, the OPE from
measured reactivity was about 21% higher on average. The value was 30% higher at
Heshan site under similar assumptions. This percentage was close to the percentage of
missing reactivity, indicating the ignorance of unmeasured or unknown organic
species can cause significant underestimation in ozone production calculation. When
the four branched-alkenes were included in the OPE calculation, the OPE results
would be 4% higher than the OPE constrained by calculated reactivity, but still far
from the OPE constrained by measured reactivity.

Compared to other similar calculations worldwide, the OPE results for Beijing

and Heshan were significantly higher (Fig 14). The comparison was made for $NO_x =$
20 ppbV which was in the range of most observation results. For urban measurements,
only the results from Mexico City in MCMA-03 were close to the Beijing results in
basic model run (Lei et al., 2008). For suburban measurements, the OPE in Heshan
2014 was higher than all other three campaigns, even including the results from
Shangdianzi station in CAREBEIJING-2008 campaigns (Ge et al., 2012). While
taking missing reactivity into consideration, the OPE results were even higher,
indicating more ozone was produced by the reactions of the same quantity of $NO_x$
molecules.

## 5. Conclusions

In this study, total OH reactivity measurements employing CRM system were
conducted at PKU site in Beijing 2013, and Heshan site 2014 in PRD region.
Comparisons between measured and calculated, as well as modelled reactivity were
made and possible reasons for the missing reactivity have been investigated. The
contribution of missing reactivity to ozone production efficiency was evaluated.
In Beijing 2013, daily median result for measured total OH reactivity was $20 \pm$
$11$ s$^{-1}$. Similar diurnal variation with other urban measurements was found with peaks
over 25 s$^{-1}$ during the morning rush hour and lower reactivity than 16 s$^{-1}$ in the
afternoon. In Heshan 2014, total OH reactivity was $31 \pm 20$ s$^{-1}$ on daily median result.
The diurnal variation was not significant. Both sites have experienced OH reactivity
over 80 s$^{-1}$ during polluted episodes.
Missing reactivity was found at both sites. While in Beijing the missing
reactivity made up 21% of measured reactivity, some periods even reached a higher
missing percentage as 40%-50%. In Heshan, missing reactivity's contribution to total
OH reactivity was 32% on average and quite stable for the whole day. Unmeasured
primary species, such as branched-alkenes could be important contributor to the
missing reactivity in Beijing, especially in morning rush hour, but they were not
enough to explain Aug 17[th] morning's event. With the help of RACM2, unmeasured
secondary products were calculated and thus the modelled reactivity could agree with
measured reactivity in Beijing in the noontime. However, they were still not enough
to explain the missing reactivity in Heshan, even in daytime. This was probably
because of the relatively higher oxidation stage in Heshan than in Beijing.
Missing reactivity could impact the estimation of atmospheric ozone production
efficiency. Compared to modeled reactivity from base run, ozone production
efficiency would rise 21% and 30% in Beijing and Heshan with measured reactivity
applied. Both results were significantly higher than similar observations worldwide,
indicating the relatively faster ozone production at both sites.
However, in order to further explore the OH reactivity in both regions, more
efforts should be paid in both OH reactivity measurements and speciated
measurements, as well as modeling to close the total OH reactivity budget. Moreover,
a thorough way with more detailed mechanisms should be established to connect the
missing reactivity to the evaluation of ozone production.

## Acknowledgement


This study was funded by the National Key Research and Development Plan (grant no.
2016YFC020200), Natural Science Foundation for Outstanding Young Scholars
(grant no. 41125018) and a Natural Science Foundation key project (grant
no.411330635). The research was also supported by the European Commission
Partnership with China on Space Data (PANDA project). Special thanks to Jing Zheng,
Mei Li, Yuhan Liu from Peking University and Tao Zhang from Guangdong
Environmental Monitoring Center for the help, thanks for William. C. Kuster from
NOAA . Earth System Research Laboratory for the branched-alkenes data in 2005.

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

Table 1 Total OH reactivity measurements in urban areas

| Campaign | Site | Year | method | $k_{OH(measured)}$ (s⁻¹) [a] | $k_{OH\ (calculated)}$ (s⁻¹ if it is a value) [b] | Measured species [c] | Reference |
|---|---|---|---|---|---|---|---|
| SOS | Nashville, US | summer, 1999 | LIF-flow tube | 11.3 | 7.2 | SFO | Kovacs et al., 2001; 2003; |
| PMTACS-NY 2001 | NY, US | summer, 2001 | LIF-flow tube | 15~25 | within 10% | SFO | Ren et al., 2003 |
| PMTACS-NY 2004 | NY, US | winter, 2004 | LIF-flow tube | 18-35 | statistically lower | SF | Ren et al., 2006a |
| MCMA-2003 | Mexico City, Mexico | spring, 2003 | LIF-flow tube | 10~120 | 30% less than | -[d] | Shirley et al., 2006 |
| TexAQS | Houston, US | summer, 2000 | LIF-flow tube | 7~12 | agree well | SFO | Mao et al., 2010 |
| TRAMP2006 | Houston, US | summer, 2006 | LIF-flow tube | 9-22 | agree well | SFOB | Mao et al., 2010 |
|  | Tokyo, Japan | 2003-2004 | LP-LIF | 10~100 | 30% less than | SFOB | Sadanaga et al., 2004; Yoshino et al., 2006 |
|  | Tokyo, Japan | summer, 2006 | LP-LIF | 10~55 | 30% less than | SFOB | Chatani et al., 2009 |
|  | Tokyo, Japan | spring, 2009 | LP-LIF | 10~35 | 22% less than | SFOB | Kato et al., 2011 |
|  | Tokyo, Japan | winter, 2007, autumn, 2009 | LP-LIF | 10~80 | 10~15 less than | SFOB | Yoshino et al., 2012 |

Table 1 Total OH reactivity measurements in urban areas (continued)

| | | | | | | | |
|---|---|---|---|---|---|---|---|
| | Mainz, German | summer, 2005 | CRM | 10.4 | | - | Sinha et al.,2008 |
| MEGAPOLI | Paris, France | winter, 2010 | CRM | 10~130 | 10~54% less than | SO | Dolgorouky et al., 2012 |
| ClearfLo | London, England | summer, 2012 | LP-LIF | 10-116 | 20~40% | SFOB | Whalley et al., 2016 |
| | Lille, France | autumn , 2012 | CRM, LP-LIF | ~70 | Reasonable agreement | SFO | Hansen et al., 2015 |
| | Dunkirk, France | summer, 2014 | CRM | 10-130 | | - | Michoud et al., 2015 |

a. For sources from different studies, the measured reactivity was presented as the averaged results, or ranges of diurnal variations, or the ranges of the whole campaign.

b. For sources of different studies, the calculated reactivity was presented within an uncertainty range, as a percentage reduction or s$^{-1}$ reduction.

c. Measured species that have been used for the calculated reactivity (following Lou et al., 2010): S = inorganic compounds (CO, NO$_x$, SO$_2$ etc) plus hydrocarbons (including isoprene); F = formaldehyde; O = OVOCs other than formaldehyde; B = BVOCs other than isoprene;

d. "-" means a lack of information regarding what has been measured or how long it has been measured.













Table2 Total OH reactivity measurements in suburban and surrounding areas

| Campaign | Site | Year | method | $k_{OH(measured)}$ ($s^{-1}$) | $k_{OH\ (calculated)}$ ($s^{-1}$ if it is a value) | Measured Species | Reference |
|---|---|---|---|---|---|---|---|
| | Central Pennsylvania, US | spring, 2002 | LIF-flow tube | 6.1 | | - | Ren et al., 2005 |
| PMTACS-NY2002 | Whiteface Mountain, US | summer, 2002 | LIF-flow tube | 5.6 | within 10% | - | Ren et al., 2006b |
| TORCH-2 | Weybourne, England | spring, 2004 | LIF-flow tube | 4.85 | 2.95 | SFO | Ingham et al., 2009; Lee et al., 2010 |
| CareBeijing-2006 | Yufa, China | summer, 2006 | LP-LIF | 10-30 | agree well | S | Lu et al., 2010; 2013 |
| PRIDE-PRD | Backgarden, China | summer, 2006 | LP-LIF | 10~120 | 50% less than | S | Lou et al., 2010 |
| DOMINO | El Arenosillo, Spain | winter, 2008 | CRM | 6.3~85 | | SF | Sinha et al., 2012 |
| | spring, 2013 | | CRM | 53 | 23 | SFOB | Kumar & Sinha., 2014 |

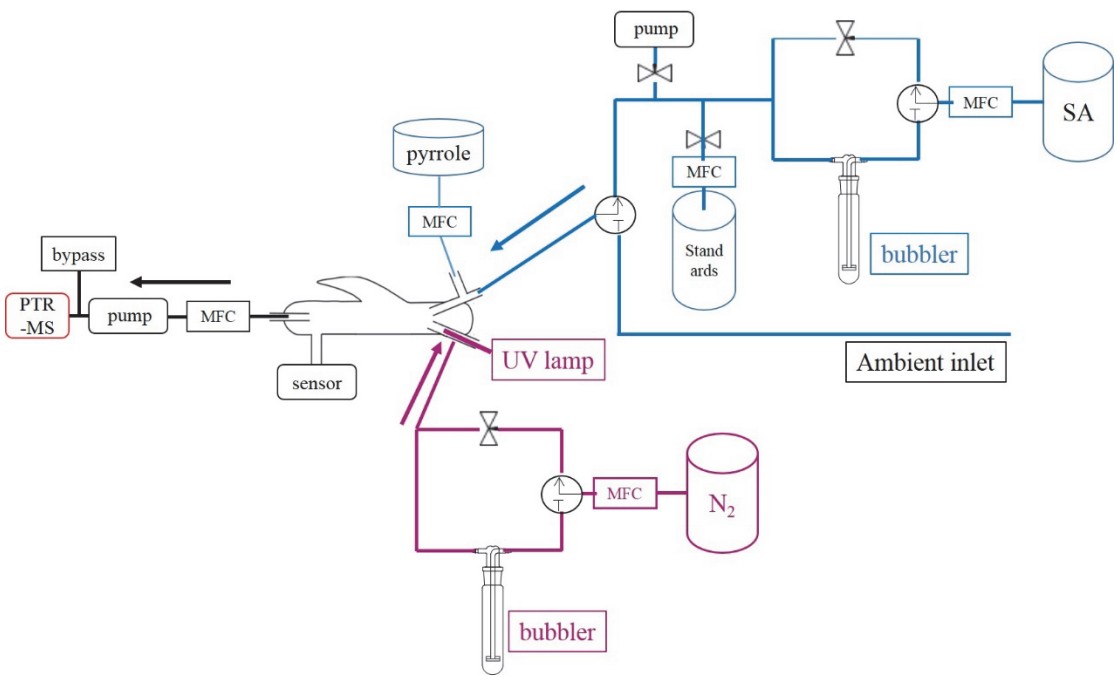


Fig 1 Schematic figures of CRM system in Beijing and Heshan observations.
Blue color represents ambient air or synthetic air injection system, purple color
represents OH generating system, black color represents the detection system.
Pressure is measured by the sensor connected to the glass reaction.

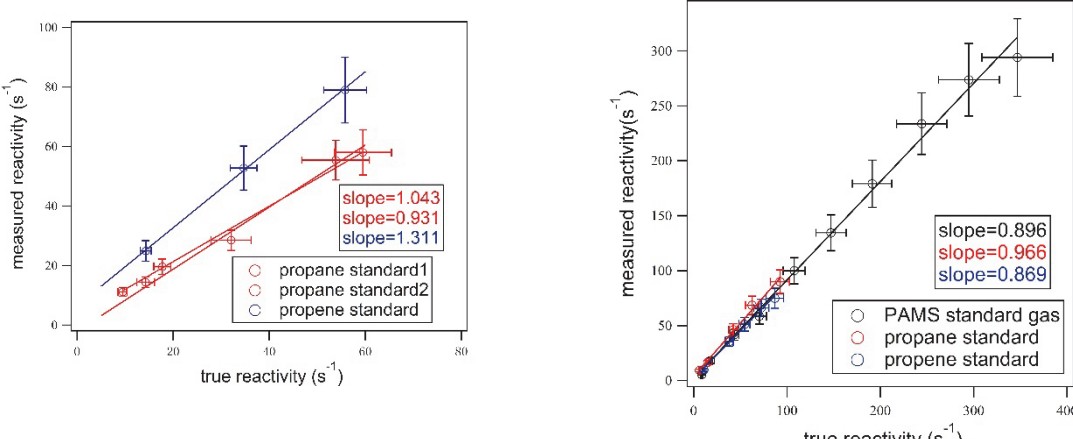

Fig 2 OH reactivity calibration in Beijing (left) and Heshan (right).
Left: Calibration in Beijing used two single standards: propane, propene;
Right: Calibration in Heshan used three standards: propane, propene, mixed PAMS 56
NMHCs.
Error bars stand for estimated uncertainty on the measured and true reactivity.

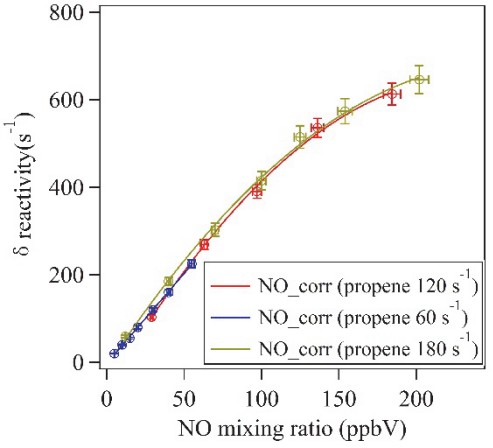 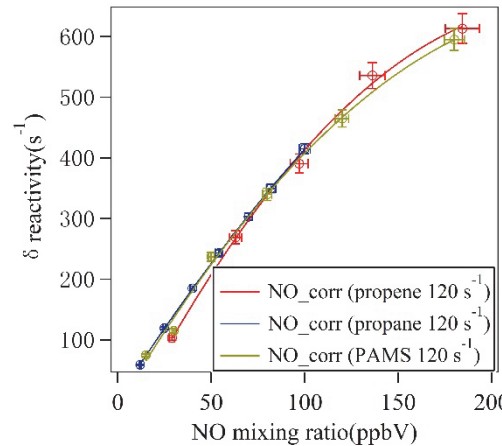


Fig 3 NO-correction experiments and fitting curves in Heshan 2014.
Left: NO-correction experiments with different mixing ratios of propene standard gas;
Right: NO-correction experiments with different standard gases at the same reactivity
level: 120 s$^{-1}$.
Error bars stand for estimated uncertainty on the NO mixing ratios and difference in
reactivity.

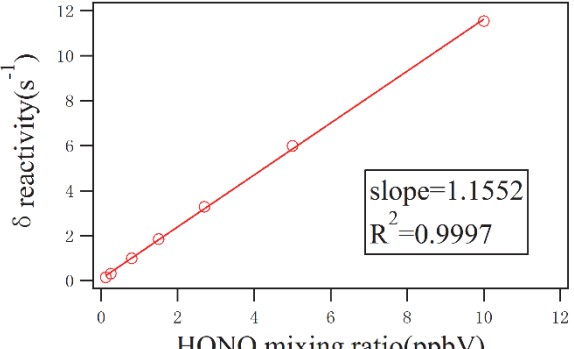


Fig 4 HONO-correction experiments and the fitting curve in Heshan 2014.


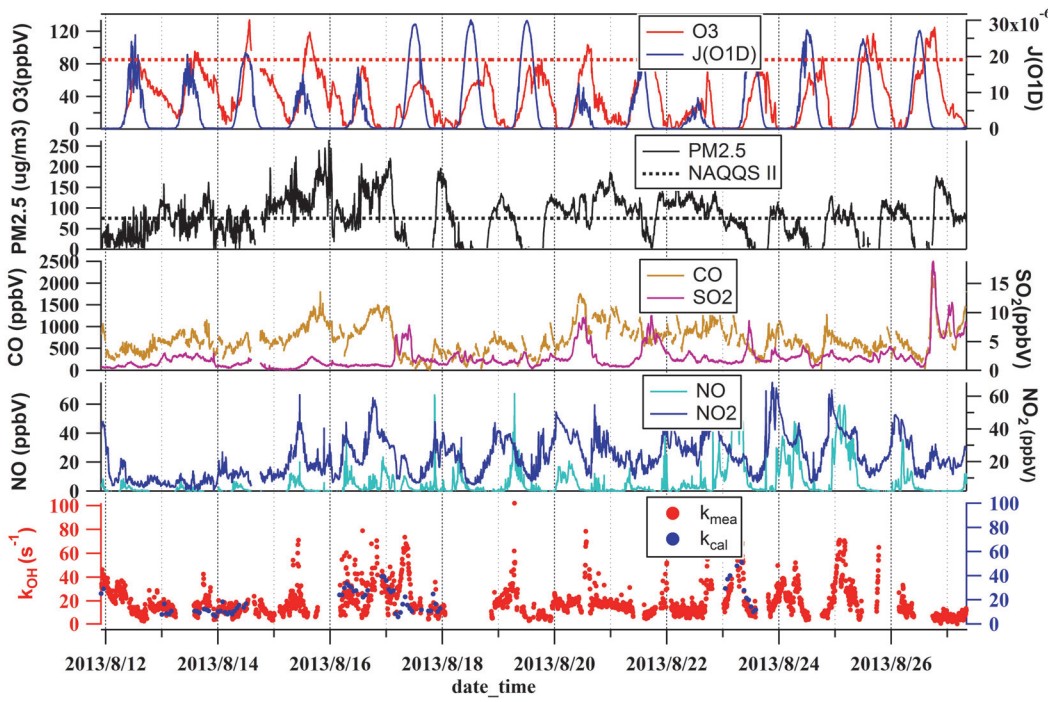

Fig 5-a Time series of meteorological parameters and inorganic trace gases during
August 2013 in Beijing.
Red and black dashed lines are Grade II of National Ambient Air Quality Standard.

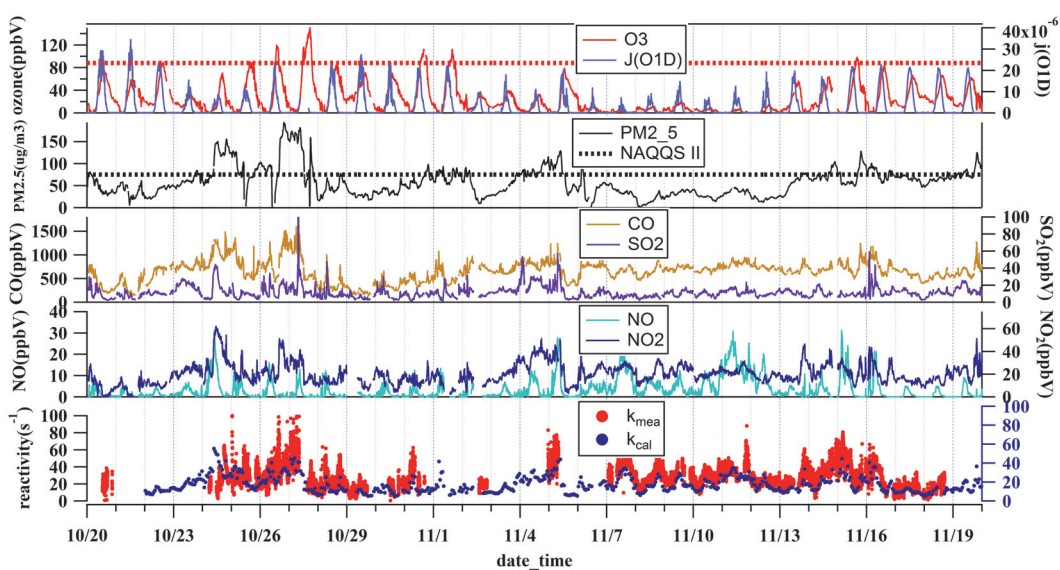

Fig 5-b Time series of meteorological parameters and inorganic trace gases during
October-November, 2014 in Heshan.
Red and black dashed lines are Grade II of National Ambient Air Quality Standard.


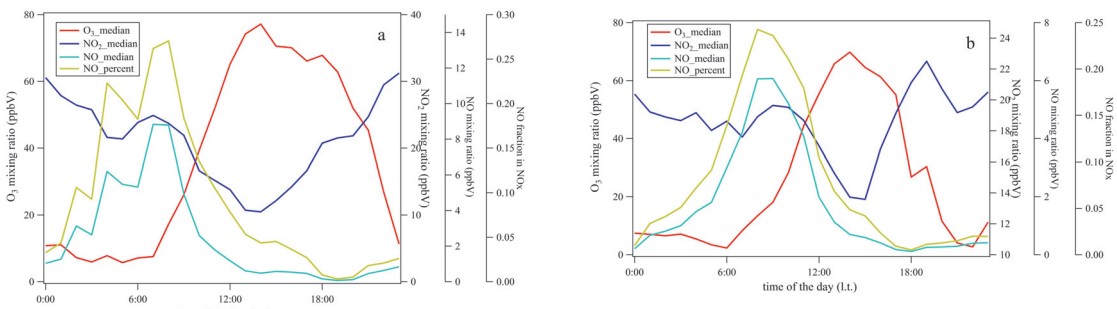

Fig 6 Diurnal variations of $O_3$, NO, $NO_2$ and relative contribution of NO to $NO_x$
in Beijing 2013 (a) and Heshan 2014 (b)

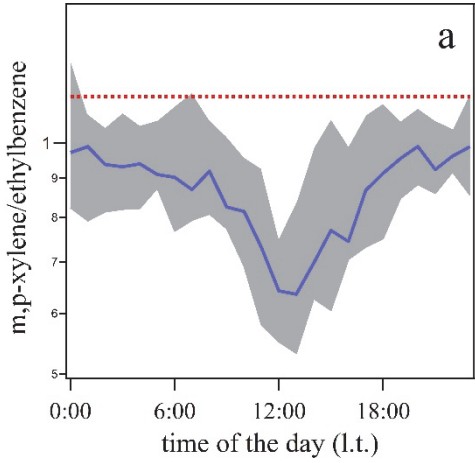
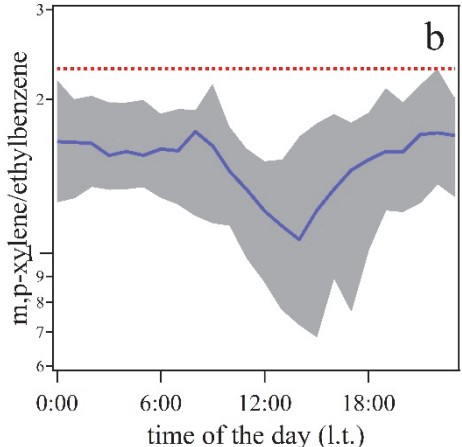


Fig 7 Ratios of m,p-xylene to ethylbenzene in Beijing 2013 (a) and Heshan 2014 (b)
Red dots line: the highest m,p-xylene to ethylbenzene ratio, assumed as emission
ratios of m,p-xylene to ethylbenzene, 1.15 ppbV ppbV$^{-1}$ in Beijing 2013 (a) and 2.3
ppbV ppbV$^{-1}$ in Heshan 2014 (b).
Shaded regions: Standard deviation for the ratios during the campaign average.

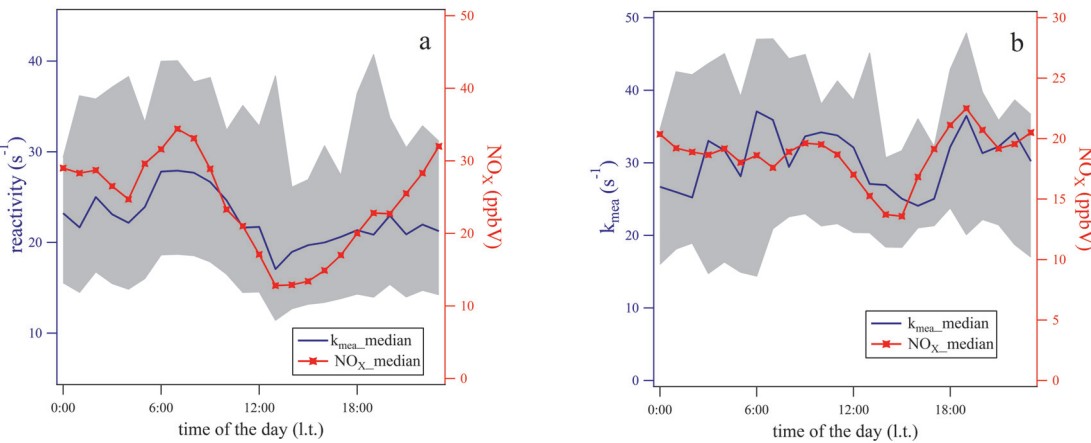


Fig 8 Diurnal variation of hourly median results of measured OH reactivity and $NO_x$
mixing ratios in Beijing (a) and Heshan (b)

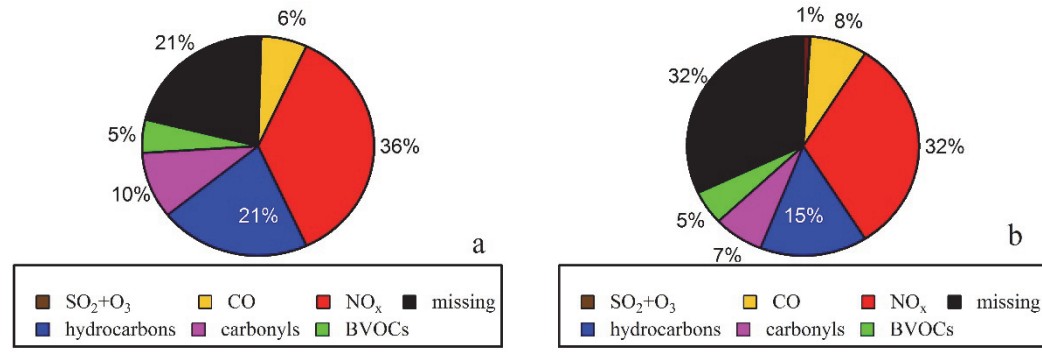

Fig 9 Composition of measured reactivity in Beijing (a) and Heshan (b)

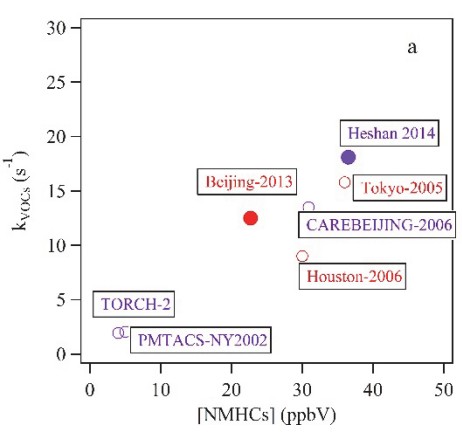
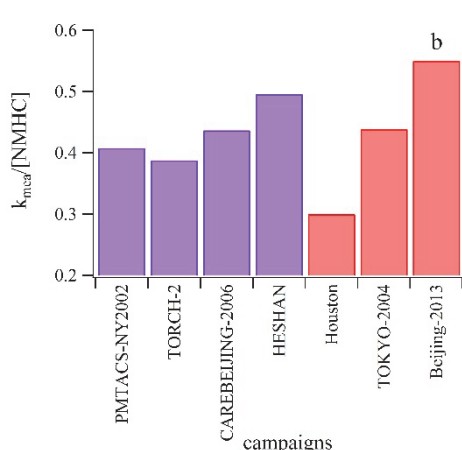


Fig 10 a: Comparison of VOCs reactivity and measured NMHCs in urban and
suburban observations.
b: Comparison of the ratio between VOCs reactivity and measured NMHCs in urban
and suburban observations

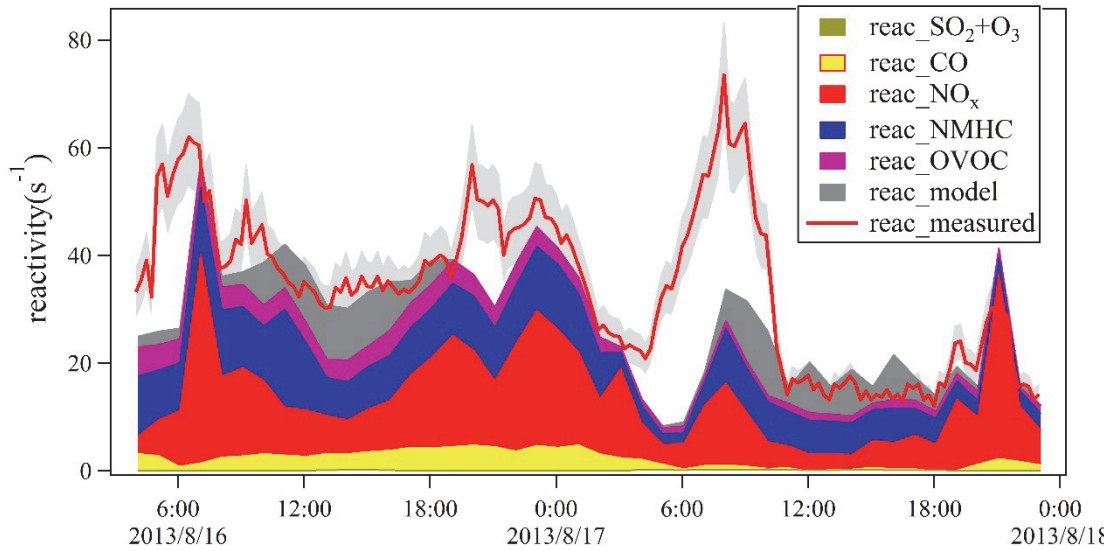

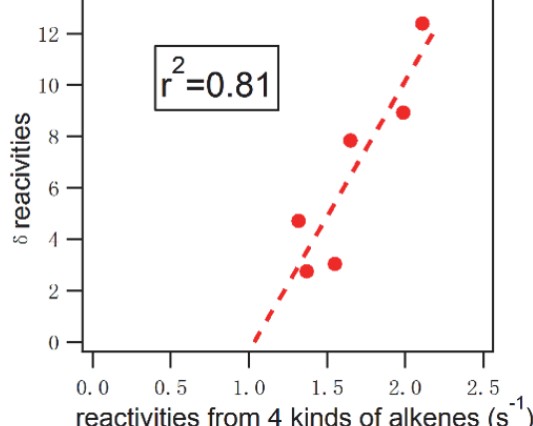

Fig 11 Upper panel: Comparison between measured and calculated reactivity in Beijing August 16[th] to 18[th] 2013.

Shallow shaded region: uncertainty of measured reactivity. The same shallow shaded region in Fig 12 represents the same.

Lower panel: Correlation between missing reactivity measured in 2013 and reactivity assumed from branched-chain alkenes from 2005 in diurnal patterns. The 4 branched-alkenes are iso-butene, 2-methyl-1-butene, 3-methyl-1-butene and 2-methyl-2-butene.


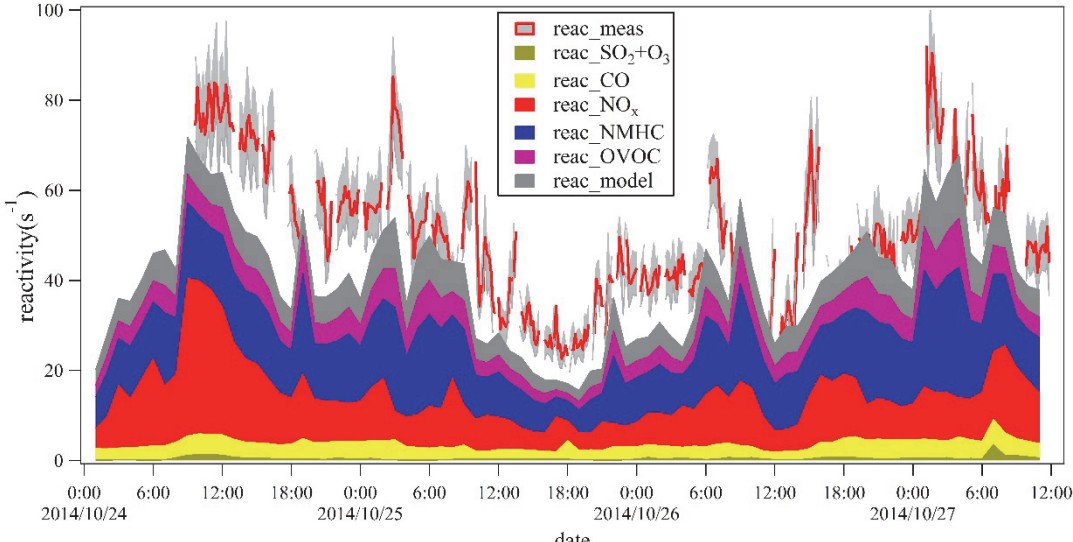

Fig 12 Comparison between measured reactivity and calculated reactivity as well as modelled reactivity in Heshan between October 24th and 27th 2014.


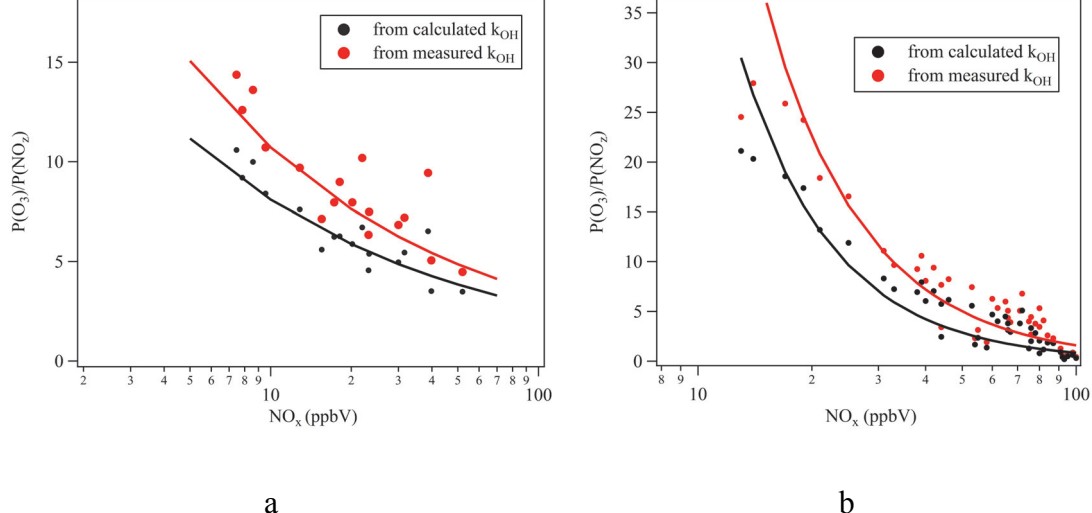


a           b

Fig 13 Comparison between OPE calculated from measured reactivity and calculated reactivity in Beijing (a) and Heshan (b).


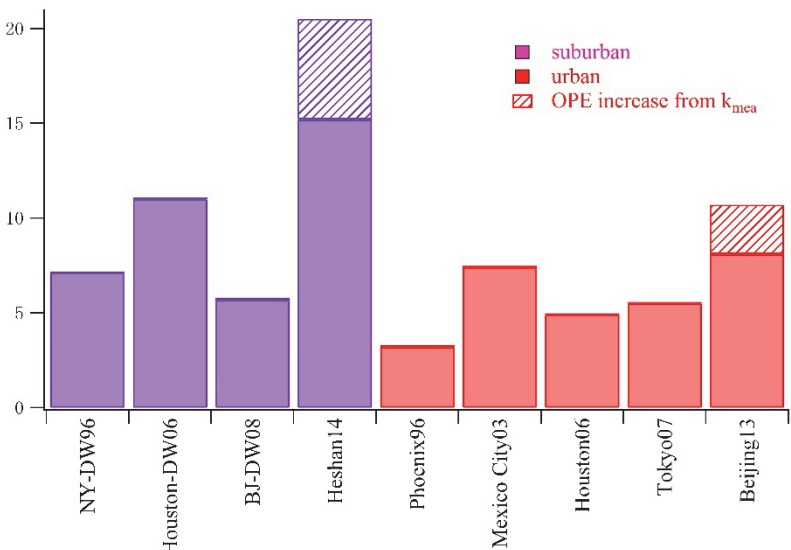


Fig 14 Comparison between the OPE results in this study and other results from literatures. The comparison is made with the $NO_x = 20$ ppbV. "DW" is in abbreviation of downwind.
