# Peer review of "How does the OH reactivity affect the ozone production efficiency"

_Atmospheric Chemistry and Physics, 2016_

## Referee Comment (RC1) · Anonymous Referee #1 · 7 Aug 2016

**General Comments:**

The paper by Yang et al. reports CRM OH reactivity measurements conducted in two Chinese cities : Beijing in August 2013 over a 17 day period and in Heshan between 19 Oct – 22 Nov 2014. Using the measured datasets and a zero dimensional box model, the authors investigate the source of missing OH reactivity in Beijing (21% ) and Heshan (32%) and also calculate the ozone production efficiency (defined as ratio of ozone production rate/ NOx production rate). They conclude that the ozone production efficiencies (OPE) would be significantly underestimated at both sites (by 27% in Beijing and 35% in Heshan), if the OPE were not constrained by the measured OH reactivity. Hence they conclude that OH reactivity measurements are necessary for accurate determination of ozone production efficiencies.

The paper is interesting and has attempted deployment of the CRM technique in very challenging high NOx concentration environments. Such a study would certainly be of great interest to the ACP readership from the perspective of fundamental process based understanding of OH reactivity and ozone production at two important sites in China. The efforts of the authors ought to be appreciated from this perspective and the study could be a valuable addition to the literature. However, there are some major technical concerns concerning the quality of the measurements and analyses, which need to be clarified/addressed/corrected by the authors before one can have confidence in the dataset and conclusions. The presentation and language also needs to be improved before it can be considered suitable for publication in ACP.

**Major Concerns:**

1) **OH reactivity measurements:** The authors have provided a good qualitative description of the CRM measurement system, but this description is generic for the CRM technique. In order to assess the quality of the measurements, relevant technical information pertaining to the operating conditions must also be provided. For example: What was the pyrrole /OH concentration ratio inside the CRM reactor during these measurements? Did it change between the deployments? What were the typical pyrrole concentrations for the C2 , C0 and C1 stages? What was the dilution factor

for ambient air inside the reactor (i.e. flow of syn air/amb air to total flow)? What was the residence time of air inside the reactor?

It appears that numerical simulations for calculating deviations from the first order conditions were neither carried out nor applied. Why? While it is good that the authors tested the accuracy of the system using propene, propane and a hydrocarbon mixture, it can be seen from the results (Figure 2) that the slopes obtained are rather different for propene (1.31) Vs propane (0.93 to 1.04). If the pyrrole (C1 concentration) /OH (C1-C2 concentration) ratio was different in these calibration experiments, it could explain the same and then correcting for the deviations from first order conditions for the relevant pyrr/OH ratio, would ensure better accuracy and take into account correction factors for each type of standard.

**2) NO-correction experiments:** Figure 3: The results are rather strange. From the paper it is not clear whether the NO concentration shown on the x axis are the ambient NO concentrations or the NO concentration inside the CRM reactor after dilution. The authors state that they have used the former (Lines 177-179; Page 6). This would be inappropriate to use considering the non linearity of NOx-VOC chemistry for OH formation.  Instead corrections are valid only for NO concentrations inside the reactor.

I am also puzzled  that the correction factor (delta $s^{-1}$ on y axis) is the same in magnitude for 60 $s^{-1}$ of OH reactivity as well 120 $s^{-1}$ of OH reactivity at the same NO concentration. The NO correction should logically be lower at 120 $s^{-1}$, as the additional reactivity would compete with NO more efficiently for the $HO_2$ resulting in lowering the OH formation due to NO+ $HO_2$. A simple numerical simulation would reveal the same. At the large NO concentrations observed during their deployments (> 10 ppb NO), it is difficult to trust that the highly non linear secondary chemistry effects arising from NO+ RO2  and NO + HO2 or even HONO photolysis inside the reactor can be accurately corrected. Note that the secondary chemistry is also occurring in a mixture containing ambient air that has several reactive compounds that have different chemistry from the standard mixtures used and some of the compounds are even unknown. In such scenario, relying on calibrations involving just propane or propene or a mixture of few

compounds cannot yield robust correction factors at high NO concentrations. For the high NO concentrations and OH reactivity conditions encountered in the present study, the authors should have used the flexibility of the CRM method in terms of adjusting the dilution factor of ambient air inside the CRM reactor. They could have then ensured that the NO concentrations were at most few ppb inside the CRM reactor, keeping the effect of NO induced secondary chemistry and its magnitude much lower. In the context of the present study, applying corrections that maybe at times 100% or even higher in magnitude relative to the uncorrected OH reactivity measurements is rather disconcerting.  It renders most of the reported OH reactivity a function of the correction factor! This effect is apparent in Fig 5 a and 5 b, when one looks at the correlation between the time series of NO and the corrected measured OH reactivity shown in the graph. The authors need to discuss this issue more comprehensively. In this regard, correlation plots of the measured OH reactivity (y axis) Vs the NO concentration (x axis) and the missing OH reactivity Vs NO concentration at both sites using the temporal data (5or even 10 min averages would do), would help much to shed light on the above point. Also, it would be good to add the time series of calculated OH reactivity to Fig 5 a and 5 b for an idea of temporal variations in both the measured OH reactivities and calculated OH reactivities. At present only few days of time series data from both sites have been presented in Fig 11 and Fig 12.

The authors may also wish to consider excluding the ambient OH reactivity data for periods when the NO concentrations lead to corrections of the same magnitude as the uncorrected OH reactivity data (100% or more). In that case, the present findings of unexplained OH reactivity may need to be revisited. At the very least, suggestions on how to perform the CRM measurements better in high NOx environments can be an outcome of the present study.

2) **$NO_2$ measurements and ozone production efficiency calculation in model using calculation $NO_2$:** The authors mention (Lines 221-223; page 8) that the $NO_x$ measurements were performed using the chemi-luminescence technique (Instrument  Model 42i, Thermo Fischer Inc., U.S.). If this is the case then obviously their $NO_2$ measurements would suffer from

a positive bias (see for e.g. Reed et al. 2016; Atmos. Chem. Phys., 16, 4707–4724, 2016) due to interference from organic nitrates, nitrous acid (which the authors report was quite high during their study without showing the actual temporal data) and nitric acid. The magnitude of such differences in inter-comparison studies involving more specific $NO_2$ instruments have been shown to overestimate the $NO_2$ concentration by upto 400% during the daytime (see Chapter 2 by Kleffman et al. in Disposal of Dangerous Chemicals in Urban Areas and Mega Cities, NATO Science for Peace and Security Series C: Environmental Security, I. Barnes and K.J. Rudzin´ski (eds.), DOI 10.1007/978-94-007-5034-0_2, # Springer Science+Business Media Dordrecht 2013. The authors mention that they used the measured $NO_2$ to constrain the model (lines 246-248; page 9). If the measured NO2 is significantly in error (say 50%), how would it affect the results of their analysis keeping in mind the implications for the unexplained OH reactivity and the calculation of the ozone production efficiency using $P(NO_z)$ (Equation 2-3). This could affect their conclusions majorly and the authors should address this concern.

**OTHER COMMENTS:**

Abstract:

(Line 22; Page 1) and elsewhere in the paper: The authors should report the measured OH reactivities and other concentration measurements by rounding off to significant figures.

(Line 34; Page 2): English is wrong: "….was presumably attribute to oxidized species…."

Introduction:

Equation 1-1: The concentration is expressed as small $[x_i]$. Please replace by capital $X_i$ as in subscript here and elsewhere.

Line 46, page 2: should be "reactive" instead of "reductive"

Lines 50-61: The authors should include other more recent relevant measurements of OH reactivity from another suburban site in Asia in Table 2, reported by Kumar, V et al., Int. J. of Mass Spectrom., 374, 55-63, 2014.

Line 108; Page 4: Although Yang et al. 2016 is cited, Sinha et al., 2012 were the first to outline this approach and use it for determining ozone production regimes.

Line 125-126; Page 5: Mention the inlet residence time. What sorts of inlet filters were used? How often were they changed under such polluted conditions?

Line 159-164; Page 6: Humidity adjustments to synthetic air for matching ambient humidity changes: It is difficult to imagine how the simple needle valve contraption can dynamically track and adjust to the ambient humidity. Some data showing the m39 water cluster concentrations of the PTR-MS for the C2 and C3 stages would be helpful.

Line 167; page 6: Spelling error: "genrated"

Lines 185-201: HONO interference: The authors do not show or mention anywhere the actual HONO concentration measurements. Considering that the OH concentration inside the CRM reactor is typically several tens of ppb, it is difficult to understand how typical ambient HONO concentrations of few hundred ppt can cause a significant interference, through the mechanism outlined by the authors. At this stage I am beginning to wonder if the authors have a fair assessment of what goes on inside the CRM reactor.

In general, the figures, their captions and legends need to be improved.

In the supplement Tables S3 & S4 are identical except for the captions!

In the supplement Tables S5 and S6 are identical except for the captions!

Section 2.3.1: How is the dilution accounted for in the model? What is the model estimated concentration of PAN and its production rate? The latter would have bearing on their assumption of approximating P (NOz) as P (HNO3)

Section 3.1: lines 270-273, Figure 5-a: During some periods (e.g. evening of 17-08-2013 and 24-08-2013) CO mixing ratios are close to zero in Beijing. How can one have trust in such measurements?

Section 3.1: Figure 5-a: There seems to negative mass concentration of PM2.5 on 17, 19, 23, 25 and 26 October evening which are masked because the axis starts from zero. Please explain.

Figure 6 : I cannot make out much as the it is difficult to read as the Figure legends are hardly legible. There does not seem to be much difference between the peak NO/NO2 ratios between Beijing and Heshan. Then how can one be sure that Heshan is influenced more strongly by transported air masses ( see Lines 301-303)? Please clarify.

L 317 and 319: The unit of OH concentration should be molecules cm$^{-3}$ (and not mole cm$^{-3}$). Similarly in L314 and 316, mole should be replaced by molecules.

Figure 7 a and b: Please mention what the red dotted lines and shaded regions signify?

Figure 8: Use appropriate legends? Not clear what is meant by "reac_mea_median"

L338-340: For Heshan, another pollution episode can be observed on 5th November despite lower CO, NOx and other pollutants. Please mention that too.

L348-352: Reasoning is not clear:
This contradicts the previous statement that Heshan receives more aged airmasses . Aged airmasses should also show more contribution from OVOCs. Moreover if faster photo-chemical production during August in Beijing, is really the reason, then one would expect less contribution from primary hydrocarbons in Beijing and more contribution of primary hydrocarbons in Heshan.

L 425: Please provide the major contributors among unmeasured aldehydes.
L437 and Table S5 and S6: Please provide full name of "ALD", GLY+MGLY", "ISOP" and "DCP" at-least once. As mentioned in the text, HCHO is a measured species. Please provide the value of $k_{cal}$ due to HCHO in table S5 and S6 also for comparison.

Section 4.3: Please mention the range of OPE or an average OPE in the text also to provide an estimate of number of ozone molecules produced per molecule of NOx consumed.

Lines 464-469: The argument is circular:
First the authors assume that the missing reactivity is due to alkenes and OVOCs which have high ozone production potential. Next the model is constrained by the "assumed" species in the second scenario. Obviously these species will provide extra ozone production.

The LOD reported for the CO, $NO_x$ and $O_3$ instruments in the supplement are much less than what is claimed by the instrument manufacturers in their manuals (40ppt for $NO_x$, 1ppb for $O_3$ and 40ppb for the 48i trace level enhanced CO analyzer). Please provide the details of how you obtained a lower LOD for these instruments or correct the Table in the supplement.

---

## Referee Comment (RC2) · Anonymous Referee #2 · 10 Aug 2016

The manuscript presents OH reactivity measurements from two urban sites in China and compares the OH reactivity data to calculated and modelled reactivity determined from the individually measured, co-observed OH sinks. Ozone production efficiency (OPE) is calculated from measured and modelled reactivity and the authors conclude that missing OH reactivity can increase ozone production efficiency at both sites. Understanding total OH reactivity by considering the dominant species contributing to OH reactivity and identifying missing OH reactivity and how this influences ozone production in urban environments is important and a suitable subject for ACP. The conclusion that more aged air-masses have a higher % of missing reactivity is an important finding also. Unfortunately there are several major problems with the manuscript currently

which mean that the results and interpretation of the results are over-shadowed: The technical concerns (already thoroughly covered by Reviewer 1) relating to the quality of the OH reactivity data from the CRM instrument under high NOx conditions where large corrections have been applied need to be addressed before final publication. Furthermore, a more comprehensive comparison between the observed reactivity and calculated and modelled reactivity should be included and discussed to strengthen the overall conclusions drawn. I struggled to evaluate much of the discussion and conclusions, largely due to the poor English, but also because data discussed in the text did not appear in the referred figures or tables: the modelled reactivity is not included in Fig. 11 upper panel and the breakdown of modelled reactivity is duplicated in tables S5 and S6. The figure axes and figure and table captions are inadequate to understand the data presented and there are inconsistencies between the data presented in the figures and discussion provided in the text. I have made a number of recommendations below where further clarification is needed or where the discussion should be improved before final publication can be considered.

Line 30: '..by adding unmeasured oxygenated..' this suggests that the model was constrained to assumed concentrations of OVOCs, but I don't think this was the case so this sentence needs revising

Line 34: change '..such as aldehydes..' to '..such as unmeasured aldehydes..'

Model description: Line 230: How were the VOC data inputted into the model given the 1 hour time resolution of these measurements and 5 min time resolution of the model?

In section 4 the authors consider the contribution unmeasured primary and secondary VOCs may make to missing reactivity. To strengthen this discussion some commentary is needed on the sensitivity of modelled OH reactivity to some of the assumed model parameters: Line 251: Are there local sources, e.g. roads, which mean that unconstrained products are not in steady state? How different is modelled reactivity on day 1 vs day 3 spin up? Lines 251-253: How sensitive is modelled OH reactivity to the

treatment of dry deposition in both locations? How was the changing boundary layer height treated in the model? Could this influence the diurnal profile of the modelled OH reactivity?

Results: Lines 317 and 319: Are these the peak OH concentrations at both sites? Given the photochemical age of the air masses wouldn't a mean OH concentration be most appropriate for this calculation? The authors should discuss briefly the sensitivity of the photochemical age to [OH] used.

Line 348: Please provide details on the data used to generate the pie-charts – is this the campaign average picture? How does this change in Heshan during the pollution episodes? It would be informative to include a time-series of calculated OH reactivity, modelled OH reactivity and measured OH for the whole of the two campaign periods somewhere in the manuscript.

Line 349: '..more significant role..' give % contributions.

Line 356-357: What was the level of the NO correction applied to the measurement data during morning rush-hour?

Discussion: Line 363: what is meant by 'relative reactivity'?

Line 366: '..not very high..' apart from Paris, Heshan VOC reactivity is highest. This section needs to be revised to accurately reflect the data in Fig. 10.

Figure 10a: Why not change the x axis to calculated NMHC reactivity (s-1)? This would then help to demonstrate the cause for this trend, i.e. a) that the type of measured NMHCs in Beijing are indeed more reactive with respect to OH than at other sites or b) missing reactivity is more significant in Beijing and Heshan vs other sites. The discussion provided in 4.1 should be revised once this figure is changed.

Figure 10b: The y axis label is missing. Also why is Paris not included in this plot?

Line 400: It is unclear whether the NOAA 2005 dataset is from Beijing? Even if it

is, it doesn't seem reasonable to simply compare missing reactivity from 2013 with branched alkene data from 2005. Could a common species, measured both in 2005 and 2013, which is strongly correlated to the branched alkene data be used to scale the 2013 branched alkene data? Why are there only 6 pts in figure 11, lower panel?

Figure 11 upper panel: modelled reactivity needs to be added to this plot

Lines 424 - 426: Key to ozone control strategies, the authors should discuss the primary species from which the modelled species derive.

Table S5: Species names should be provided in full – what is 'DCB'?

Lines 444-445: the authors should also compare the calculated and modelled reactivity from 2006 and 2014 too, so the 50% higher measured reactivity in 2014 can be evaluated fully.

Lines 452, 453: 'PAMS 56 hydrocarbons' and 'T0-15 OVOCs' need defining

Lines 465-466: in section 4.3 the authors report that the OH reactivity modelled in Beijing agreed with measured reactivity in the daytime (lines 423-424), but on lines 465-466 report differences between measured and modelled reactivity in Beijing which changed OPE by 27%. These two statements are inconsistent with each other and as the modelled reactivity is missing from Figure 11 it is unclear which is correct.

---

## Referee Comment (RC3) · Anonymous Referee #3 · 14 Aug 2016

Review of Atmos. Chem. Phys. Manuscript (#acp-2016-507) "How does the OH reactivity affect the ozone production efficiency: case studies in Beijing and Heshan" by Y. Yang et al.

**General Comments**

This paper presents OH reactivity measurements in Beijing and Heshan, China using the CRM technique. Missing OH reactivity was found in both studies and its impact on ozone production efficiency was assessed using a box model. The scope of this work important and can improve our understanding of ozone production and I think this work is worth publishing. My big concern is the uncertainty in the OH reactivity measurement, which may reduce the significance of missing OH reactivity on the ozone production. Another concern is that in the ozone production efficiency (OPE) calculation, the NOx loss due to organic nitrate formation is omitted, i.e., only the formation of nitric acid from OH + NO2 is considered. This could over-estimate the OPE depending on the relative importance of organic nitrate production over nitric acid production in these two environments. Overall I found the manuscript needs much improvement in English. The authors need to clarify many things in several parts of the manuscript (see Special Comments below), especially the discussion of the effect of the measurement uncertainty on importance of missing OH reactivity and the clarification on how the measurement corrections was done due to interference of humidity and NO. In addition, I would ask the authors to consider the following special comments in their revision.

**Special Comments**

1. L.2: The country should be added, i.e. "Case studies in Beijing and Heshan, China"
2. L.22-23: with a detection limit of 5 s^-1 stated in L.198, it is not possible that two decimals in the OH reactivity values can be significant. Integer numbers probably enough.
3. L.23: it should read "Measurements in Beijing presented …"
4. L.25-26: need to define missing OH reactivity here. I can guess it is the different between measured OH reactivity and OH reactivity calculated from measured OH reactants. If so, state so.
5. L.32: "However, the model failed to explain the missing reactivity in Heshan,", but was the box model able to explain the missing OH reactivity in Beijing?
6. L.35-36: it should read "…when the model is constrained by the measured reactivity…".
7. L.48: remove "researches" or change it to "calculations".
8. L.70: "…the 75% missing reactivity in Paris in MEGAPOLI under continental air masses influences", need a reference for this statement. It should be mentioned that missing OH reactivity in each study depends heavily on the completeness of measurement suite, especially VOC species, so the next paragraph can follow.
9. L.89-90: change "…in one case could increase reactivity by over 50%..." to "…in one case which could increase reactivity by over 50%...".
10. L.91: an increasing concern.
11. L. 92: What is the ozone level for Grade II of China National Ambient Air Quality Standards?
12. L.94-95: "…it appears there is an increasing trend for ozone in Beijing and other area…" for what time frame? Recent years?

13. L.104: change those to which.
14. L.106: from the total OH reactivity.
15. L.109: change "…two intensive observation datasets conducted…" to "…data from two intensive field studies conducted…"
16. L.114-116: consider to change this sentence to: "The possible missing reactivity and its importance for the ozone production calculation are discussed."
17. L.125-126: consider to change this to: "a 14.9 m Teflon inlet with an outer (I assume) diameter of 3/8 inch…".
18. L.136: Some impurities in dry air and nitrogen could also be photolyzed.
19. L.142-143: total ambient (?) OH reactivity is calculated as…
20. In Fig. 2, the color for symbols with different standards is not clearly shown. Consider to use different symbols and/or change to different colors for symbols and lines with a better color contrast.
21. L.155-156: What is the "uncertainty range for all calibrations"?
22. L.157-165 about correction due to humidity: it is not clear how exactly this correction was done based on Figure S2, where no labels for x and y axes are given so we really do not know what is plotted here. If the pyrrole signal versus relative humidity are plotted, why there are negative values?
23. L.166-182 about the correction due to NO: in Fig. 3 the y axis is labeled as delta reactivity. Is this the difference between calculated (standards) and measured OH reactivity? In the legends of the figure, there are reactivity numbers (60/120/18 s^-1) and I assume these are based on the OH reactivity calculated from the contents in the standard gases. If this is correct, why can the delta reactivity be 300-600 s^-1? As stated in L.173-174, the "measured" reactivity decreased as the NO mixing ratio increased. If so, the measured OH reactivity should be lower than the calculated values and the difference should be also less than these numbers (60/120/18 s^-1). Please clarify.
24. L.192-193: Was the correction associated with HONO interference also applied to the measurements in both sites, or to the measuremens in Heshan only, since it looks like there is no HONO measurements in Beijing from Table S1.
25. L.198: 2$\sigma$ instead of 2$\delta$.
26. L198-201: is the uncertainty of 20% for 1$\sigma$ or 2$\sigma$? Shouldn't the uncertainty associated with NO correction be taken into account?
27. L.248-251: it is not clear what output results from the box model was used in the calculations, time dependent results or stead-state result? The authors mentioned both a time-dependent mode of 5 min and stead-state conditions with a 3-day spin-up time. Please clarify.
28. L.263: P(NOz) should also include the production rates of organic nitrates which can be calculated using the box model results. Depending the fraction of organic production rate in the total NOx consumption rate, the OPE could be significantly over-estimated.
29. L.272: include units for the measurement results in Heshan. Are the errors standard deviations?
30. L.279: (O$^1$D).
31. L.281: is 93 ppbv for hourly or 8-hour maximum?
32. L.282-284: How come that VOC concentrations in Beijing and Heshan are the same (i.e., Table S3 and Table S4 are identical)? Alkanes made up over 60% of the summed VOCs in Beijing. Is this in terms of concentration or VOC reactivity? Please clarify.

33. L.294: photochemical age is mentioned here but it is not defined until next paragraph. It's not presented in Fig. 6-7 either.
34. L.314: please define LTC.
35. L.325: again the two decimals are not significant considering the relatively large uncertain of 5 s^-1 in the measurements. Please correct all reported numbers for OH reactivity.
36. L.330-331: the morning rush hour peak could be because of a shallow boundary layer.
37. L.335-338: I don't think the little variations in OH reactivity on clean day can fully explain the less variability of OH reactivity in Heshan and in Beijing. It's probably because the air sampled in Heshan is more aged (as the authors have discussed) and regionally mixed than in Beijing.
38. L.353-354: please give the absolute missing OH reactivity values in s^-1 for both location. A comparison between the missing OH reactivity and the combined uncertainty of measured (5 s^-1) and calculated (from the measured species) OH reactivity is needed in order to see if the missing OH reactivity is significant. The uncertainty of measured OH reactivity should be also discussed somewhere in Section 4 when the contribution to the missing reactivity is discussed.
39. L.357: the entire campaign.
40. L.363-364: "the relative reactivity compared to NMHCs mixing ratios were higher." Higher than what?
41. L.399: the sentence, "We found only one dataset in 2005 measured by NOAA (Liu et al., 2009)." is not clear to me. One dataset of what?
42. L.400-402 and Fig, 11 lower panel: missing OH reactivity is plotted against the reactivity assumed from 4 branched alkenes. What are these 4 alkenes; are they representative for the missing alkenes; and how is the calculation performed? Please clarify.
43. L.402-403: consider to change this to: "even with the mixing ratios of the 4 branched alkenes measured in Beijing in 2005, the reactivity…"
44. L419-420: need a reference for the statement that the mixing ratios of branched alkenes could be lower than 0.1 ppbv. The site in Beijing is only a few hundred meters from major roads and can easily get influenced by vehicle emissions.
45. L.425-426 and Table S5: not sure if I understand the "major secondary contributors to modeled reactivity. Why only these species are listed in Table S5? Are they related unmeasured intermediates that are calculated in the model? Need definitions for the acronyms in the model (ALD, DCB, etc.).
46. L.460-462: It's not clear to me how the remaining missing reactivity were allocated into different intermediates. Are these intermediates constrained (remain constant) in the box model run in the second scenario?
47. L.466: on average. Also please give absolute values, i.e., increase from XX to YY.
48. Section 4.4: again, the OPE needs to be recalculated by including the production rate of organic nitrates in P(NOz). This may change the picture currently shown in Fig. 14.
49. L.497-499: Need to include the modeled OH reactivity in Fig. 11. Without this, it is hard to assess this statement that missing OH reactivity can be reconciled with modeled intermediates that were not measured. Also this statement seems in contrast with the statement in L.395-399, where the author stated that unmeasured primary VOCs, especially branched alkenes, are responsible for the missing OH reactivity. Please clarify.

50. L.507-510: probably add a sentence stating that efforts to reduce the uncertainty of OH reactivity measurements based on the CRM technique to increase the confidence of results as shown in this work.
51. Fig. 5: please give the Grade II of National Ambient Air Quality Standard for ozone and PM2.5 in the caption. Also it seems the two red lines in Fig. 5-a and Fig. 5-b are different, one above 80ppbv and the other below 80 ppbv. Please clarify.
52. Fig. 6: the yellow (or brown) lines show the NO fraction (not percentage) in NOx. Please correct it.
53. Fig. 11: please plot the modeled OH reactivity, the same as in Fig. 12. Is the gray area along the red line showing the uncertainty of the measurement? If so, please state this in the caption.
54. Fig. 14: references for OPEs in other studies should be given.
55. Again, Table S5 and Table S6 are identical.
56. P.3 of the supplement: in Alkenes, ethane should be ethene.

---

## Author Comment (AC1) · 22 Jan 2017

**Reply to Reviewer's comments**

*General Comments*

The paper by Yang et al. report CRM OH reactivity measurements conducted in two Chinese cities: Beijing in August 2013 over a 17 day period and in Heshan between 19 Oct – 22 Nov 2014. Using the measured datasets and a zero dimensional box model, the authors investigate the source of missing OH reactivity in Beijing (21%) and Heshan (32%) and also calculated the ozone production efficiency (defined as ratio of ozone production rate/NOx production rate). They conclude that the ozone production efficiencies (OPE) would be significantly underestimated at both sites (by 27% in Beijing and 35% in Heshan), if the OPE were not constrained by the measured OH reactivity. Hence they conclude that OH reactivity measurements are necessary for accurate determination of ozone production efficiencies.

The paper is interesting and has attempted deployment of the CRM technique in very challenging high NOx concentration environments. Such a study would certainly be of great interest to the ACP leadership from the perspective of fundamental process based understanding of OH reactivity and ozone production at two important sites in China. The efforts of the authors ought to be appreciated from this perspective and the study could be a valuable addition to the literature. However, there are some major technical concerns concerning the quality of the measurements and analyses, which need to be clarified/addressed/corrected by the authors before one can have confidence in the datasets and conclusions. The presentation and language also needs to be improved before it can be considered suitable for publication in ACP.

Response: Appreciate your comments. As you have said above, there're a lot of problems in our previous manuscript, whether in technical part or language part. Wish these answers and latest version of the paper could help with it.

*Major Concerns:*
*1) OH reactivity measurements: The authors have provided a good qualitative description of the CRM measurement system, but this description is generic for the CRM technique. In order to assess the quality of the measurements, relevant technical information pertaining to the operating conditions must also be provided. For example: What was the pyrrole/OH concentration ratio inside the CRM reactor during these measurements? Did it change between the deployments? What were the typical pyrrole concentrations for the C2, C0 and C1 stages? What was the dilution factor for ambient air inside the reactor (i.e. flow of syn air/amb air to total flow)? What was the residence time of air inside the reactor?*

*It appears that numerical simulations for calculating deviations from the first order conditions were neither carried out nor applied. Why? While it is good that the authors tested the accuracy of the system using propene, propane and a hydrocarbon mixture, it can be seen from the results (Figure 2) that the slopes obtained are rather different from propene (1.31) Vs propane (0.93 to 1.04). If the pyrrole (C1 concentration)/OH*

*(C1-C2 concentration) ratio was different in these calibration experiments, it could explain the same and then correcting for the deviations from first order conditions for the relevant pyrr/OH ratio, would ensure better accuracy and take into account correction factors for each type of standard.*

Response: Thanks for asking. This is a mistake and in the latest version of manuscript, we have included some fundamental parameters of the CRM system. In the last paragraph of 2.1.1, we re-phrase it as below:
Ambient air or synthetic air was introduced at 160 -170 ml min$^{-1}$ with the total flow 320 – 350 ml min$^{-1}$(The typical dilution factor was about 2-2.15 depending on the situation). The residence time of air inside the reactor was less than 30 s before they were pumped by the Teflon pump. The typical c1 mixing ratio for pyrrole in Beijing and Heshan measurements were about 60 ppbV and 55 ppbV, while the mixing ratios of OH radicals generated by mercury lamp were about 35 ppbV and 28 ppbV. The mixing ratios were quite consistent for either of the campaigns, respectively.
Also we have tried the FACSIMILE model for the deviation to achieve the equations 2-2 and 2-3.
Also your suggestion may be right, the pyrrole/OH ratios from propene and propane were different due to the slightly different ratios used in two periods of time. This would introduce uncertainty in our measurements.

*2) NO-correction experiments: Figure 3: The results are rather strange. From the paper it is not clear whether the NO concentration shown on the x axis are the ambient NO concentrations or the NO concentrations inside the CRM reactor after dilution. The authors state that they have used the former (Lines 177-179; Page 6). This would be inappropriate to use considering the non-linearity of NOx-VOC chemistry for OH formation. Instead corrections are valid only for NO concentrations inside the reactor.*

*I am also puzzled that the concentration factor (delta s-1 on y axis) is the same in magnitude for 60 s-1 of OH reactivity as well 120 s-1 of OH reactivity at the same NO concentration. The NO correction should logically be lower at 120 s-1, as the additional reactivity would compete with NO more efficiently for the HO2 resulting in lowering the OH formation due to NO+HO2. A simple numerical simulation would reveal the same. At the large NO concentrations observed during their deployment (>10 ppb NO), it is difficult to trust that the highly non-linear secondary chemistry effects arising from NO+RO2 and NO+HO2 or even HONO photolysis inside the reactor can be accurately corrected. Note that the secondary chemistry is also occurring in a mixture containing ambient air that has several reactive compounds that have different chemistry from the standard mixtures used and some of the compounds are even unknown. In such scenario, relying on calibrations involving just propane and propene or a mixture of few compounds cannot yield robust corrections factors at high NO concentrations. For the high NO concentrations and OH reactivity conditions encountered in the present study, the authors should have used the flexibility of the CRM*

*method in terms of adjusting the dilution factor of ambient air inside the CRM reactor. They could have then ensured that the NO concentrations were at more few ppb inside the CRM reactor, keeping the effect of NO induced secondary chemistry and its magnitude much lower. In the context of the present study, applying corrections that maybe at times 100% or even higher in magnitude relative to the uncorrected OH reactivity measurements is rather disconcerting. It renders most of the reported OH reactivity a function of the correction factor! This effect is apparent in Fig 5a and 5b, when one looks at the correlation between the time series of NO and the corrected measured OH reactivity shown in the graph. The authors need to discuss this issure more comprehensively. In this regard, correlation plots of the measured OH reactivity (y axis) Vs the NO concentration (x axis) and the missing OH reactivity Vs NO concentration at both sites using the temporal data (5 or even 10 min averages would do), would help much to shed light on the above point. Also, it would be good to add the time series of calculated OH reactivity to Fig 5a and 5b for an idea of temporal variations in both the measured OH reactivity and calculated OH reactivity. At present only few days of time series data from both sites have been presented in Fig 11 and Fig12.*

*The authors may also wish to consider excluding the ambient OH reactivity data for periods when the NO concentrations lead to corrections of the same magnitude as the uncorrected OH reactivity (100% or more). In that case, the present findings of unexplained OH reactivity may need to be revisited. At the very least, suggestions on how to perform the CRM experiments better in high NOx environments can be an outcome of the present study.*

Response: Thanks for the advice. This is the key question to the CRM technique in this work. For the first question, the NO concentrations we used in the calculation were "ambient" NO concentrations, which were introduced into the system with synthetic air. We assumed this was the starting situation for the reaction and calculation. We did not have the NO measurements in the CRM reactor and could not quantify the concentrations well, thus we could not calculate the coefficients for the equations.

The second question remains unsolved even to us. We would expect the same situation as the referee did. The difference in quantities of reactants should have a different influence on the NO recycling and thus a different delta reactivity. However, after we've tried the experiments several times with different standard gases, we've found the correction curve were very close. We are still pursuing the principles underlying but we've not reached a persuasive one.

For the method, most of the colleagues suggest that we should deploy the advantages of flow adjustment. However, that advantage is very useful when the observation site is changed or the forecasts predict the coming of a heavy polluted air masses. However, like in Peking University Site, which was close to vehicular emissions, high levels of NO always come in several minutes and not enough time is left to flow adjustment and system stabilization. If we set the ratio of nitrogen to ambient air too high, maybe the OH reactivity of diluted samples will be too low to reach detection limits. But with our experiments, we believe in our corrections and employ the correction curve to obtain

the measured reactivity. And for these two observations, the corrections in the manuscript presented the best we can do at that time.

For the relationship between measured reactivity and NO, we found similar diurnal patterns between measured reactivity and $NO_x$, which were dominated by $NO_2$ in both sites. For the comparison between measured and calculated reactivity, we would like to focus on the missing reactivity in some pollution episodes. However, if the referee insists, we could supply with the similar figures as Fig 5a and Fig 5b with calculated reactivity included.

Thank you very much for these three paragraphs discussion. NO interference is really the most important and annoying problem concerning CRM technique. For further researches, more effort will be put in the experimental and analyzing work. For experimental part, the experiments should be undertaken more carefully and systematically to figure out the key factors influencing the NO interference. For analytic part, we need to find one way to find the correction curve and explain it in a principle way.

*3) $NO_2$ measurements and ozone production efficiency calculation in model using calculation $NO_2$: The authors mention (Lines 221-223; page 8) that the $NO_x$ measurements were performed using the chemi-luminescence technique (Instrument Model 42i, Thermo Fischer Inc., U. S.). If this is the case then obviously their NO2 measurements would suffer from a positive bias (see for e.g. Reed et al. 2016; Atmos. Chem. Phys., 16, 4707-4724, 2016) due to interference from organic nitrates, nitrous acid (which the authors report was quite high during their study without showing the actual temporal data) and nitric acid. The magnitude of such differences in inter-comparison studies involving more specific $NO_2$ instruments have been shown to overestimate the $NO_2$ concentration by up to 400% during the daytime (see Chapter 2 by Kleffman et al. in Disposal of Dangerous Chemicals in Urban Areas and Mega Cities, NATO Science for Peace and Security Series C: Environmental Security, I. Barnes and K. J. Rudzin Ski (eds,), DOI 10.1007/978-94-007-5034-0_2, # Springer Science + Business Media Dordrecht 2013. The authors mention that they used that measured NO2 to constrain the model (lines 246-248; page 9). If the measured NO2 is significantly in error (say 50%), how would it affect the results of their analysis keeping in mind the implications for the unexplained OH reactivity and the calculation of the ozone production efficiency using $P(NO_z)$ (Equation 2-3). This could affect their conclusions majorly and the authors should address their concern.*

Response: Thanks for the advice. This recent piece of literature really brings a big question about the reliability of $NO_2$ measurements. Especially in our calculation and analysis, the contribution from $NO_2$ to total OH reactivity is very large. However, we could only supply with one piece of supportive evidence for our manuscript. In Heshan, we have two set of $NO_x$ analyzers. One was equipped with a home-build photolytic converter ($NO_x$-PL) and the other one was equipped with a catalytic converter ($NO_x$-Mo), both were similar as the setups of the instruments in Tan et al (2017). This comparison could help us to figure how great the interference could $NO_z$ have on the

measurements. From the data comparison, we found that the datasets were close within 10% for most of the time but could be over 20% when in some nights or morning time. However, we have no similar inter-comparison in Beijing measurements. However, this could be a great interference and should call our attentions in later research.

*Other Comments:*
*Abstract:*
*(Line 22; Page1) and elsewhere in the paper: The authors should report the measured OH reactivities and other concentration measurements by rounding off to significant figures.*

Response: Thanks for the advice. Modified as suggested.

*(Line 34; Page 2): English is wrong: "… was presumably attribute to oxidized species…"*

Response: Sorry for this mistake and revised as suggested.

*Introduction:*
*Equation 1-1: The concentration is expressed as small $[x_i]$. Please replace by capital $X_i$ as in subscript here and elsewhere.*

Response: Thanks for the advice and changed as suggested.

*Line 46, Page2: should be "reactive" instead of "reductive"*

Response: Thanks for asking. However, we think the species here are the species could react with OH radicals, which should be reductive species. Appreciate your suggestion anyway for this clarification.

*Lines 50-61: The authors should include other more recent relevant measurements of OH reactivity from another suburban site in Asia in Table 2, report by Kumar, V et al., Int. J. of Mass. Spectrom., 374, 55-63, 2014.*

Response: Thanks for the advice about this latest results and included in Table 2 in last manuscript.

*Line 108; Page 4: Although Yang et al. 2016 is cited, Sinha et al., 2012 were the first to outline this approach and use it for determining ozone production regimes.*

Response: Thanks for advice. Modified as suggested.

*Line 125-126: Page 5: Mention the inlet residence time. What sorts of inlet filters were used? How often were they changed under such polluted conditions?*

Response: Thanks for these details. The last sentence of this paragraph was modified as below: Ambient air was sampled after a teflon filter and then pumped through a 14.9m Teflon 3/8 inch (outer diameter) inlet at about 7 L·min-1 rate, with a 5 - 6 s residence time.

*Line 159-164; Page 6: Humidity adjustments to synthetic air for matching ambient humidity changes: It is difficult to imagine how the simple need valve contraption can dynamically track and adjust to the ambient humidity. Some data showing the m39 water cluster concentrations of the PTR-MS for the C2 and C3 stages would be helpful.*

Response: Thanks for asking. This is really an important part for the work. The needle valve we used in the experiments could not adjust itself to certain humidity. However, once we set it to certain position and it could remain the relative humidity for quite a long time. However, as it was tuned manually rather than automatically, it could not track the ambient humidity and could only be adjusted by the operator when he was free to check the humidity and adjust it. That was also why the correction equation was needed for further corrections. The figure for one example is supplied in latest version of supplement information.

*Line 167; Page6: Spelling error: "generated"*

Response: Thanks for the careful check and modified as suggested.

*Lines 185-201: HONO interference: The authors do not show or mention anywhere the actual HONO concentration measurements. Considering that the OH concentration inside the CRM reactor is typically several tens of ppb, it is difficult to understand how typical ambient HONO concentrations of few hundred ppt can cause a significant interference, through the mechanism outlined by the authors. At this stage I am beginning to wonder if the authors have a fair assessment of what goes on inside the CRM reactor.*

Response: Thanks for asking. First, HONO results ranged from several tens of pptV to several ppbV in Heshan campaign. However, the data was under-preparation for publications now so I did not present the figures in this paper. Second, while the HONO was just 100 pptV or so, the influence was not very significant. However when the ambient HONO mixing ratio was over 5 ppbV, which was observed in the campaign, the influence should not be omitted. Moreover, the HONO peak came out several times with low reactivity periods, which called for more careful corrections.

*In general, the figures, their captions and legends need to be improved.*

Response: Thanks for suggestions and modified.

*In the supplement Table S3 and S4 are identical except for the captions!*
*In the supplement Tables S5 and S6 are identical except for the captions!*

Response: Thanks for the check and very sorry for this careless mistake. Modified in the new manuscript.

*Section 2.3.1: How is the dilution accounted for in the model? What is the model estimated concentration of PAN and its production rate? The latter would have bearing on their assumptions of approximating P(NOz) as P(HNO3)*

Response: Thanks for this question very much. Referee 3 also brought this question, which was a big problem in previous calculation. In the new version, we include the organic nitrate part in the calculation and the results could be a little different.

*Section 3.1: lines 270-273, Figure 5-a: During some periods (e.g. evening of 17-08-2013 and 24-08-2013) CO mixing ratios are close to zero in Beijing. How can one have trust in such measurements?*

Response: Thanks for asking. We've checked the data. There're some points too low which we think the data was influenced by the zero-calibration. However, for most of the data, we think they're valid. I think some of the points appeared very low in the figure due to the large axis. When the CO mixing ratio was about 100 ppbV, you could not find the point in the figure. However, the suggestion is important for us.

*Section 3.1: Figure 5-a: There seems to negative mass concentration of PM2.5 on 17, 19, 23, 25 and 26 October evening which are masked because the axis starts from zero. Please explain.*

Response: Thanks for asking. We're using the TEOM for PM2.5 measurements, and our instrument could not measure precisely when the PM2.5 concentration was very low. So there're some points below zero but that makes no sense so we delete the data below zero and then we got the figure as you saw.

*Figures 6: I cannot make our much as the it is difficult to read as the Figure legends are hardly legible. There does not seem to be much difference between the peak NO/NO2 ratios between Beijing and Heshan. Then how can one be sure that Heshan is influenced more strongly by transported air masses (see Lines 301-303)? Please clarify.*

Response: Thanks for the question. The figures were modified in the latest version of manuscript. NO/NOx ratio was lower in Heshan compared to Beijing, and the other factor was the time when NO went down to near zero. With these two factors, and also other explanation listed after this paragraph, we assume that Heshan is influenced by transported air masses compared to Beijing.

*L317 and 319: The unit of OH concentration should be molecules cm-3 ( and not moel cm-3). Similarly in L314 and 316, mole should be replaced by molecules.*

Response: Thanks for the advice and revised as suggested.

*Figure 7a and b: Please mention what the red dotted lines and shaded regions signify?*

Response: Thanks for the suggestion and modified in the figures.

*Figure 8: Use appropriate legends? Not clear what is meant by "reac_mea_median"*

Response: Thanks for the suggestion and modified as $k_{mea}$\_median and explained as the median results of measured reactivity.

*L338-340: For Heshan, another pollution episode can be observed on 5th November despite lower CO, NOx and other pollutants. Please mention that too.*

Response: Thanks for mentioning that. This was correct but we select the two episodes for the accumulation of secondary pollutants.

*L348-352: Reasoning is not clear: This contradicts the previous statement that Heshan receives more aged air masses. Aged air masses should also show more contribution from OVOCs. Moreover if faster photochemical production during August in Beijing, is really the reason, then one would expect less contribution from primary hydrocarbons in Beijing and more contribution of primary hydrocarbons in Heshan.*

Response: Thanks for the question. I think the expression in previous manuscript was not good. So we re-phrase it with no expression comparing the OVOC percentages in both sites. Your suggestion is right that the measuring seasons were different in both sites which would significantly influence the secondary species production rate. However, there was another possibilities lying in the missing reactivity. However, the previous expression was inappropriate. Appreciate the advice.

*L425: Please provide the major contributors among unmeasured aldehydes. L437 and Table S5 and S6: Please provide full name of "ALD", "GLY+MGLY", "ISOP" and "DCP" at least once. As mentioned in the text, HCHO is a measured species. Please provide the value of $k_{cal}$ due to HCHO in table S5 and S6 also for comparison.*

Response: Thanks for the advice. In the latest version, we explained the species concerning these four categories. For the HCHO part, we've included HCHO as an input for the model, so the $k_{cal}$ and $k_{mod}$ was almost the same and thus we did not present the $k_{cal}$ in the table. However, appreciate the question.

*Section 4.3: Please mention the range of OPE or an average OPE in the text also to*

*provide an estimate of number of ozone molecules produced per molecule of NOx consumed.*

Response: Thanks for the suggestions and modified as advised.

*Lines 464-469: The argument is circular: First the authors assume that the missing reactivity is due to alkenes and OVOCs which have high ozone production potential. Next the model is constrained by the "assumed" species in the second scenario. Obviously these species will provide extra ozone production.*

Response: Thanks for asking. I am not sure about the logical question here. We have mentioned two probable explanations in 4.2 and 4.3 and then take them into consideration in 4.4 to calculate the OPE here. If we exclude these alkenes and OVOCs, then the calculation would appear un-constrained. However, I am not sure whether we should take other species into consideration.

*The LOD reported for the CO, NOx and O$_3$ instruments in the supplement are much less than what is claimed by the instrument manufacturers in their manuals (40 ppt for NOx, 1 ppb for O$_3$, and 40 ppb for the 48i trace level enhanced CO analyzer). Please provide the details of how you obtained a lower LOD for these instruments or correct the Table in the supplement.*

Response: Thanks for asking. However, after checking with our engineers, these were typical running conditions for these instruments for several campaigns since CAREBEIJING-2006 and PRIDE-PRD2006 (Lu et al 2012; 2013).

---

## Author Comment (AC2) · 22 Jan 2017

**Reply to Reviewer's comments**

*General Comments*

  *The manuscript presents OH reactivity measurements from two urban sites in China and compared the OH reactivity data to calculated and modelled reactivity determined from the individually measured, co-observed OH sinks. Ozone production efficiency (OPE) is calculated from measured and modelled reactivity and the authors conclude that missing OH reactivity can increase ozone production efficiency at both sites. Understanding total OH reactivity by considering the dominant species contributing to OH reactivity and identifying missing OH reactivity and how this influence ozone production in urban environments is important and a suitable subject for ACP. The conclusion that more aged air-masses have a higher % of missing reactivity is an important finding also. Unfortunately there are several major problems with the manuscript currently which mean that the results and interpretation of the results are over-shadowed: The technical concerns (already thoroughly covered by Reviewer 1) relating to the quality of the OH reactivity data from the CRM instrument under high $NO_x$ conditions where large corrections have been applied need to be addressed before final publication. Furthermore, a more comprehensive comparison between the observed reactivity and calculated and modelled reactivity should be included and discussed to strengthen the overall conclusions drawn. I struggled to evaluate much of the discussion and conclusions, largely due to the poor English, but also because data discussed in the text did not appear in the referred figures or table: the modelled reactivity is not included in Fig. 11 upper panel and the breakdown of modelled reactivity is duplicated in tables S5 and S6. The figure axes and figure and table captions are inadequate to understand the data presented and there are inconsistencies between the data presented in the figures and discussion provided in the text. I have made a number of recommendations below where further clarification is needed or where the discussion should be improved before final publication can be considered.*

Response: Appreciate the general comments on this manuscript. We accept the comments on poor English and modified to this version. Wish this time it's easier to understand.

*Line 30: '… by adding unmeasured oxygenated…' this suggests that the model was constrained to assumed concentrations of OVOCs, but I don't think this was the case so this sentence needs revising*

Response: Thanks for asking. We are constraining the box model with several measured carbonyls, such as acetaldehyde, acetone.

*Line 34: change '.. such as aldehydes..' to '.. such as unmeasured aldehydes..'*

Response: Appreciate the suggestions and accepted.

*Model description: Line 230: How were the VOC data inputted into the model given the 1 hour time resolution of these measurements and 5 min time resolution of the model?*

Response: Thanks for asking. For VOCs data, we were using the linear interpolation method to achieve the 5 min time resolution and then put the data into model.

*In section 4 the authors consider the contribution unmeasured primary and secondary VOCs may make to missing reactivity. To strengthen this discussion some commentary is needed on the sensitivity of modelled OH reactivity to some of the assumed model parameters: Line 251: Are there local sources, e. g. roads, which mean that unconstrained products are not in steady state? How different is modelled reactivity on day 1 vs day 3 spin up? Line 251-253: How sensitivity is modelled OH reactivity to the treatment of dry deposition in both locations? How was the changing boundary layer height treated in the model? Could this influence the diurnal profile of the modelled OH reactivity?*

Response: Thanks for asking. Firstly, we admit that the model we were using in this manuscript was really a simple one. The close primary emissions, surly would introduce large uncertainty on the model job. However, as we were using this observation-based model, we could not input the emission data. The way we dealt with this problem was trying to do the moving average to avoid the sudden increase of decrease of certain species. For different spin-up time test, we found modelled OH reactivity could be different within 10% between two situations. However, for the literature suggestion and the 24 hour deposition time we chose, we decided to choose 3 day spin-up. For the dry deposition, some OVOCs and secondary species were quite sensitive to the dry deposition choice. From literature (Lu et al, 2012; 2013), we chose 24 hour dry deposition, and the modelled OVOCs presented similar diurnal variations of observations, with a 15%-30% difference depending on species. For the boundary layer height, we determined to set a well-mixed boundary layer height of about 1 km. However, this could be a source of uncertainty due to the diurnal variation of boundary layer height. Moreover, from Lu et al (2013) and Tan et al (2016), we would know that there could be missing chemistry in the nocturnal boundary layers, which would introduce unconstraint species and reactions in model work. However, due to the lack in boundary layer height measurements, we decided to set a constant boundary layer height. But the advice above are all important and appreciate all the questions.

*Results: Line 317 and 319: Are these the peak OH concentrations at both sites? Given the photochemical age of the air masses wouldn't a mean OH concentration be most appropriate for this calculations? The authors should discuss briefly the sensitivity of the photochemical age to [OH] used.*

Response: Thank you very much for this suggestion. These are both the peak OH concentrations in diurnal variations. Yes, this is one mistake and for the mean OH

concentrations should be the most appropriate. Also in the latest manuscript, we include a short sentence concerning the influence from OH concentration on photochemical age calculation. Appreciate your help.

*Line 348: Please provide details on the data used to generate the pie-charts – is this the campaign average picture? How does this change in Heshan during the pollution episodes? It would be informative to include a time series of calculated OH reactivity, modelled OH reactivity and measured OH for the whole of the two campaign periods somewhere in the manuscript.*

Response: Thanks for asking. It is the campaign average. However, it only included the data when there were OH reactivity measurement results. The data used to generate the pie-charts were as follows. However, we thought with this pie-charts, we did not need to present the data as well. For the second question, we could include the figure the referee asked for Heshan campaign. However, in Beijing observation, due to the discontinuity for OH reactivity and VOCs measurements through these three weeks, the modeled reactivity would need many times of interpolation and would thus introduce a great uncertainty here. So we decide to focus on certain processes rather than the campaign average.

*L459: '..more significant role..' give % contributions.*

Response: Thanks for asking. In latest version of manuscript, we rephrase this sentence as below: The OVOCs had also significant contribution, and measured OVOCs had a sharing of 10% in total reactivity in Beijing while 7% in Heshan. We think the comparison was influenced by many factors, so we give up this direct comparison. Appreciate for this question.

*L456-457: What was the level of the NO correction applied to the measurement data during morning rush-hour?*

Response: Thanks for asking. This is one important question. For some periods in morning rush hour, when NO mixing ratio over 20 ppbV was observed, the NO correction could be over 40 s-1 for measured reactivity. While the absolute reactivity was about the same level. However, this correction was checked for different species and verified. So we think this results were valid.

*Discussion: L363: what is meant by relative reactivity?*

Response: The relative reactivity means here the ratio between VOCs reactivity and NMHCs mixing ratios. It's not a strict definition here. However, we did not bring out this phrase at first in the latest version of manuscript. Appreciate your question.

*Line 366: '..not very high..' apart from Paris, Heshan VOC reactivity is highest. This*

*section needs to be revised to accurately reflect the data in Fig. 10.*

Response: Thanks for asking. We re-phrase this paragraph as below: The measured VOCs reactivity (obtained by subtracting inorganic reactivity from total OH reactivity), 11.2s-1 in Beijing and 18.3s-1 in Heshan (Fig 10), was actually not at high end comparing with the levels from literatures. Tokyo presented a similar level of VOCs reactivity (Yoshino et al., 2006) and Paris had an even higher level of VOCs reactivity which was obtained in wintertime (Dolgorouky et al., 2012). The measured NMHCs levels (obtained by adding all hydrocarbon mixing ratios together) were also not very high, with Beijing 2013 being around 20 ppbV and Heshan 2014 higher than 35 ppbV. The relative VOCs reactivity, defined by the ratio of the VOCs reactivity to the measured NMHCs levels, the values for both Beijing and Heshan were very high.

*Figure 10a: Why not change the x axis to calculated NMHC reactivity ($s^{-1}$) ? This would then help to demonstrate the cause for this trend, i. e. a) that the type of measured NMHCs in Beijing are indeed more reactive with respect to OH than at other sites or b) missing reactivity is more significant n Beijing and Heshan vs other sites. The discussion provided in 4.1 should be revised once the figure is changed.*

Response: Thanks for asking. Your suggestion is quite good to clarify the importance of the missing reactivity. However, if we are using the calculated NMHCs reactivity as x axis, we would not know that the first part you're telling, that the difference in the known compositions. Because we would only know that NMHCs reactivity in Beijing and Heshan were quite high. However, what we would like to express as well is while the NMHCs mixing ratios were not very high in Beijing and Heshan, the NMHCs reactivity could be high in both places. However, in our figure, there remains the questions that how could we presented both problems in one figure.

*Figure 10b: They y axis label is missing. Also why is Paris not included in this plot?*

Response: Thanks for the suggestion. Accepted and revised.

*Line 400: It is unclear whether the NOAA 2005 dataset is from Beijing. Even it is, it doesn't seem reasonable to simply compare missing reactivity from 2013 with branched alkenes data from 2005. Could a common species, measured both in 2005 and 2013, which is strongly correlated to the branch alkene data be used to scale the 2013 branched alkene data? Why are there only 6pts in figure 11, lower panel?*

Response: Thanks for asking. Yes for sure, the NOAA 2005 dataset was from Beijing. For the calculation in this part, we were firstly trying to track the correlation between the branched-alkenes and missing reactivity. So we compare the results in both diurnal variations. However, as we said in the paper. Even the mixing ratios in 2005 was not enough to explain all the missing reactivity, not even to say the decreasing trends for VOCs species in Beijing since 2005. The reason we only got 6 points were mainly due

to we would compare the data between morning rush hours in the diurnal variations. Appreciate for the questions to clarify this part.

*Figure 11 upper panel: modelled reactivity needs to be added to this plot*

Response: Accepted and modified as suggested. Thanks for the advice.

*Lines 424-426: Key to ozone control strategies, the authors should discuss the primary species from which the modelled species derive.*

Response: Thanks for asking. Sorry for this paper, the research focuses mostly on whether the primary or secondary species contributed more for the missing reactivity. However, we also have the data to answer your questions. These species were formed mostly as the oxidation products of alkenes and aromatics. However, your suggestion is very important and key to our next step – ozone control strategies. The vehicular emissions and solvent use related to alkenes and aromatics in urban areas should be controlled strictly for ozone reduction.

*Table S5: Species names should be provided in full – what is 'DCB'?*

Response: Thanks for advice. This is also the advice from other reviewers. We will give a full explanation of major species in the tables.

*Lines 444-445: the authors should also compare the calculated and modelled reactivity from 2006 and 2014 too, so the 50% higher measured reactivity in 2014 can be evaluated fully.*

Response: Thanks for the advice. However, VOCs measurements from 2006 was limited to offline canister samples and the species were fewer than 2014. We're afraid this compare may be not comprehensive.

*Lines 452,453: 'PAMS 56 hydrocarbons' and 'TO-15 OVOCs' needs defining*

Response: Accepted. We made an introduction to these two sets of standard gases, but due to the limitation of length, we would not supply the details of these species here.

*Lines 465-466:in section 4.3 the authors report that the OH reactivity modelled in Beijing agreed with the measured reactivity in the daytime (lines 423-424), but on lines 465-466 report difference between measured and modelled reactivity in Beijing which changed OPE by 27%. These two statements are inconsistent with each other and as the modelled reactivity is missing from Figure 11 it is unclear which is correct.*

Response: Thanks for asking. This is an interesting question. We've got similar questions at first. However, after we double-checked, the results remained the same

(though after we take organic nitrate into consideration, the difference was 21%). I think the difference could result from reasons below: 1) though many of the daytime reactivity were the same for the episodes, still some points the measured and modelled reactivity were different, not even to say the significant difference in rush hours; 2) the difference in species composition in scenario 1 and 2. Different species would introduce different levels of ozone production efficiency. However, this remains a question we need to dig in. Appreciate the question.

---

## Author Comment (AC3) · 22 Jan 2017

**Reply to Reviewer's comments**

*General Comments*

*This paper presents OH reactivity measurements in Beijing and Heshan, China using the CRM technique. Missing OH reactivity was found in both studies and impact on ozone production efficiency was assessed using a box model. The scope of this work is important and can improve our understanding of ozone production and I think this work is worth publishing. My big concern is the uncertainty in the OH reactivity measurement, which may reduce the significance of missing OH reactivity on the ozone production. Another concern is that in the ozone production efficiency (OPE) calculation, the $NO_x$ loss due to organic nitrate formation is omitted, i. e., only the formation of nitric acid from OH+ $NO_2$ is considered. This could overestimate the OPE depending on the relative importance of organic nitrate production over nitric acid production in these two environments. Overall I found the manuscript needs much improvement in English. The authors need to clarify many things in several parts of the manuscript (See Special Comments below), especially the discussion of the effect of the measurement uncertainty on importance of missing OH reactivity and the clarification on how the measurement corrections was done due to interference of humidity and NO. In addition, I would ask the authors to consider the following special comments in their revision.*

Response: Thank you for your commendation and appreciate for your questions and suggestion. Specially the two major concerns about the measurement uncertainty and the OPE calculation. For the measurements uncertainty, I made some clarification about the measuring setups as well as some explanation about the correction methods, following all the reviewers' comments. Hopefully this manuscript would help you to trust the validation of the method. For the OPE calculation, your advice is so helpful that we really missed the organic nitrate formation part, which would surly influence the results significantly as you have suggested. So in the new manuscript, we re-calculated the OPE and achieved new results.

Special Comments:
*1. L.2: The country should be added, i.e. "Case studies in Beijing and Heshan, China"*

Response: Accepted and modified following the suggestion.

*2. L.22-23: with a detection limits of 5 $s^{-1}$ stated in L. 198, it is not possible that two decimals in the OH reactivity values can be significant. Integer numbers probably enough.*

Response: Accepted and modified following the suggestion. However, this should be discussed, as in this method, the final results were achieved by the equation (2-1). The choice of the significant figures should be made depending on how well the experimental method can tell the difference in two close reactivity. However, this was

not discussed in this manuscript. In this one, we accept the comments.

*3. L23: it should read "Measurements in Beijing presented…"*

Response: Accepted and modified as "The data in Beijing showed".

*4. L25-26: need to define missing OH reactivity here. I can guess it is the different between measured OH reactivity and OH reactivity calculated from measured OH reactants. If so, state so.*

Response: Accepted and modified as suggested. Appreciate the suggestions.

*5. L32:"However, the model failed to explain the missing reactivity in Heshan", but was the box model able to explain the missing OH reactivity in Beijing?*

Response: The question was a good one. We can only say in the case study between August 16$^{th}$ and 18$^{th}$, 2013. The model was able to explain the missing reactivity in daytime. But for other periods or in the evening, it could not work well.

*6. L35-36: it should read "…when the model is constrained by the measured reactivity …"*

Response: Accepted and modified following the suggestion. This should be as you said. Thank you.

*7. L48: remove "researches" or change it to "calculations".*

Response: Change it to "calculations" as suggested. Appreciate that.

*8: L70: "… the 75% missing reactivity in Paris in MEGAPOLI under continental air masses influences", need a reference for this statement. It should be mentioned that missing OH reactivity in each study depends heavily on the completeness of measurement suite, especially VOCs species, so the next paragraph can follow.*

Response: Accepted and add the reference. The detail about the VOCs species measured in these campaigns are listed in table 1 and table 2. This was really important that the defined missing reactivity were really dependent on the measuring suite.

*9. L89-90 change " in one case could increase reactivity by over 50%…" to " … in one case which could increase reactivity by over 50%…"*

Response: Accepted and change to "… in once case the OVOCs could increase reactivity by over 50%..." It should be clarified here we were talking about the importance of OVOCs in reactivity. Appreciate your suggestions.

*10. L91. An increasing concern.*

Response: Accepted and change it to "Ground-level ozone pollution has been of increasing concerns in China".

*11. L92: What is the ozone level for Grade II of China National Ambient Air Quality Standards?*

Response: Sorry to fail to present the standards. So add "(93 ppbV)" after the "Grade II of China National Ambient Air Quality Standards (2012)".

*12. L94-95: "… it appears there is an increasing trend for ozone in Beijing and other area…" for what time frame? Recent years?*

Response: Appreciate the question. It was observed the increasing trend for recent years. As presented in Zhang et al, 2014, there was observed an increasing trend in one Beijing site between 2005 and 2011.

*13. L104: change those to which.*

Response: Accepted and change the sentence to "Due to the limitation of current measurement techniques, some VOCs species which could not be quantified so far, and therefore cannot be integrated into current chemical mechanisms of model run, could laid a great uncertainty in ozone production prediction".

*14. L106: from the total OH reactivity.*

Response: Accepted and change as suggested. Thanks for this advice.

*15. L109: change "… two intensive observations datasets conducted …" to "… data from two intensive field studies conducted"*

Response: Accepted and modified as suggested.

*16. L114-116: consider to change this sentence to "The possible missing reactivity and its importance for the ozone production calculation are discussed"*

Response: Accepted and modified as suggested.

*17. L124-126: consider to change this to "a 14.9 m Teflon inlet with an outer (I assume) diameter of 3/8 inch…"*

Response: Accepted and change to "Ambient air was sampled after a teflon filter and

then pumped through a 14.9m Teflon 3/8 inch (outer diameter) inlet …"

*18. L136: Some impurities in dry air and nitrogen could also be photolyzed.*

Response: This could be true. So we also did one experiment in C0 and C1 mode but without pyrrole. We observed no significant difference in m/z 68 signals. So we assume the photolysis of impurities in dry synthetic air and nitrogen would not directly cause influence on pyrrole measurements. But this question was important. Appreciate for the concern.

*19. L142-143: total ambient (?) OH reactivity is calculated as…*

Response: Accepted and change it to "total OH reactivity of ambient air …"

*20. In Fig.2, the color for symbols with different standards is not clearly shown. Consider to use different symbols and/or change to different colors for symbols and lines with a better color contrast.*

Response: Accepted and modified the figure as suggested.

*21. L155-156: What is the "uncertainty range for all calibrations"?*

Response: It means the correlation factors were within certain range taking all factors, such as the uncertainty from standard gases, the mixing into consideration.

*22. L157-165 about correction due to humidity: it is not clear how exactly this correction was done based on Figure S2, where no labels for x and y axes are given so we really do not know what is plotted here. If the pyrrole signal versus relative humidity are plotted, why there are negative values?*

Response: Sorry for the unclearness of the axis labels. The x axis represents delta m/z 37 signals, and the y axis represents delta normalized m/z 68 signals. We chose a medium relative humidity as the "zero" point. We got the $S_{37}^{0}$ and $S_{68n}^{0}$. Then we got the delta m/z 37 signals and the delta normalized m/z 68 signals as follows:
$$\text{delta m/z 37 signals} = S_{37}^{i} - S_{37}^{0}$$
$$\text{delta normalized m/z 68 signals} = S_{68n}^{i} - S_{68n}^{0}$$

*23. L166-182 about the correction due to NO: in Fig. 3 the y axis is labeled as delta reactivity. Is this the difference between calculated (standards) and measured OH reactivity? In the legends of the figure, there are reactivity numbers (60/120/180 $s^{-1}$) and I assume these are based on the reactivity calculated from the contents in the standard gases. If this is correct, why can the delta reactivity be 300 – 600 $s^{-1}$? As stated in L. 173-174, the "measured" reactivity decreased as the NO mixing ratio increased. If so, the measured OH reactivity should be lower than the calculated values and the*

*difference should be also less than these numbers (60/120/180 s $^{-1}$). Please clarify.*

Response: Thanks for the question. This was a tricky part for the explanation of "delta reactivity". The delta reactivity is defined as the difference between "measured" reactivity and "calculated" reactivity. However, this "measured" reactivity is derived from calculation of the m/z 68 signals, which equals the calculated reactivity in campaign observations. The "calculated" reactivity is actually the standard reactivity calculated from the mixing ratios of standard gases and rate coefficients, which equals the measured reactivity in campaign observations. This is a little confusing. Normally, the "measured" reactivity in NO correction experiments were lower than the "calculated" reactivity due to the OH radicals generated from NO recycling.
The delta reactivity could be even larger than the standard reactivity. This is because due to the excessive OH radicals generated from NO recycling, c3 could be even lower than c2 and negative values of "measured" reactivity could be calculated from the equation 2-1.

*24. L.192-193: Was the correction associated with HONO interference also applied to the measurements in both sites, or to the measurements in Heshan only, since it looks like there is no HONO measurements in Beijing from Table S1.*

Response: Thank you for your suggestions. Yes we only used the HONO correction in Heshan datasets. One reason is we didn't have the HONO measurements in August 2013 in Beijing. The other reason is from later measurements in Beijing, we found the HONO levels in summertime in PKU site was much lower than the results from Heshan. So we did not apply the correction in Beijing.

*25. L.198: 2σ instead of 2δ.*

Response: Appreciate your suggestion very much. Modified as suggested.

*26. L.198-201: is the uncertainty of 20% for 1σ or 2σ? Shouldn't the uncertainty associated with NO correction be taken into account?*

Response: Thanks for your question. The uncertainty of 20% is for 1 σ . This is the total uncertainty for ordinary measurements. The uncertainty associated with NO correction is largely depended on the NO mixing ratios. However in our NO correction experiments, we calculated the uncertainty between 10%-15% (1 σ ).

*27. L.248-251: it is not clear what output results from the box model was used in the calculations, time dependent results or stead-state result? The authors mentioned both a time-dependent mode of 5 min and stead-state conditions with a 3-day spin-up time. Please clarify.*

Response: Thanks for the question. For the model we were used in this work, the model

was operated in a time-dependent mode, but we need a 3-day spin-up time for warming-up. The output results were time-dependent. Appreciate for the chance to clarify.

28. L.263: P(NOz) should also include the production rates of organic nitrates which can be calculated using the box model results. Depending the fraction of organic production rate in the total NOx consumption rate, the OPE could be significantly over-estimated.

Response: Thanks for your suggestions. This is really an important one and we are sorry our previous calculation missed the organic production rate, such as organic nitrate. In this new manuscript, we include both inorganic and organic production rate.

29. L.272: include units for the measurement results in Heshan. Are the errors standard deviations?

Response: Thanks for the advice. In the latest version we include units for all results. Yes, they are standard deviations.

30. L.279: (O1D).

Response: Thank you for this detail. Modified as suggested.

31. L.281: is 93 ppbV for hourly or 8-hour maximum?

Response: Yes, it is.

32. L. 282-284: How come that VOC concentration in Beijing and Heshan are the same (i.e, Table S3 and Table S4 are identical)? Alkanes made up over 60% of the summed VOCs in Beijing. Is this in terms of concentration or VOC reactivity? Please clarify.

Response: We are so sorry for this careless mistake. We have corrected it in this new version. This percentage is in terms of volume concentration.

33. L.294: photochemical age is mentioned here but it is not defined until next paragraph. It's not presented in Fig. 6-7 either.

Response: Thanks for the advice. However, as we saw in most literatures concerning photochemical age, I found most of them considering the phrase as known to readers and present no special explanation for it. But if it's necessary, we think the equation 3-1 should be enough. Appreciate for your careful suggestion.

34. L.314: Please define LTC.

Response: Thanks for the suggestion. We have modified it to "local time".

*35. L.325: again the two decimals are not significant considering the relatively large uncertainty of 5 s⁻¹ in the measurements. Please correct all reported numbers for OH reactivity.*

Response: Thanks for this. We have modified as you suggested in 1.

*36. L.330-331: the morning rush hour peak could be because of a shallow boundary layer.*

Response: Thank you for suggestion. Yes, the shallow boundary layer could be an important factors. However, there were two reasons we think primary emissions could be more important or more significant. One is we could find NOx and hydrocarbons connected to vehicle emissions increased faster than other secondary species. The other reason is that the highest peak occurred at 7 o'clock or 8 o'clock, while the boundary layer was not the lowest. However, we admitted the variations of boundary layer really had an important influence on reactivity results.

*37. L.335-338: I don't think the little variations in OH reactivity on clean days can fully explain the less variability of OH reactivity in Heshan and in Beijing. It's probably because the air sampled in Heshan is more aged (as the authors have discussed) and regionally mixed than in Beijing.*

Response: Thanks for suggestions. The reviewer's suggestion could be one of the reasons and we have included in the revised paper. However, the 2 periods of clean air really caused a significant influence on the diurnal variation average.

*38. L.353-354: please give the absolute missing OH reactivity values in s⁻¹ for both location. A comparison between the missing OH reactivity and the combined uncertainty of measured (5 s⁻¹) and calculated (from the measured species) OH reactivity is needed in order to see if the missing OH reactivity is significant. The uncertainty of measured OH reactivity should be also discussed somewhere in Section 4 when the contribution to the missing reactivity is discussed.*

Response: Thanks for the advice. The absolute missing OH reactivity values have been included in the latest manuscript: over 4 s⁻¹ in Beijing and 10 s⁻¹ in Heshan. The direct comparison with the average missing reactivity and combined uncertainty of measured and calculated reactivity were weakened because of the difference with time-series. In some time, the measured and calculated reactivity could be close within 5 s⁻¹ difference, while in other periods like in Section 4, the difference could be over 30 s⁻¹. For this work, we focus mostly on the periods with significant missing reactivity, as picked in Section 4.

*39. L.357: the entire campaign.*

Response: Thanks for the advice. Accepted and modified as suggested.

40. L.363-364: "the relative reactivity compared to NMHCs mixing ratios were higher."

Response: Thanks for the suggestion. Accepted and modified.

41. L.399: the sentence, "We found only one datasets in 2005 measured by NOAA (Liu et al, 2009)." is not clear to me. One dataset of what?

Response: Thanks for asking and sorry for the misunderstanding. It should be the dataset of branch-alkenes measurements.

42. L.400-402 and Fig.11 lower panel: missing OH reactivity is plotted against the reactivity assumed from 4 branched alkenes. What are these 4 alkenes; are they representative for the missing alkenes; and how is the calculation performed? Please clarify.

Response: Thanks for asking. The four branched alkenes were iso-butene, 2-methyl-1-butene, 3-methyl-1-butene and 2-methy-2-butene. For the calculation, we have two datasets, one is the missing OH reactivity diurnal variation in 2013, and the other is the branched-alkenes measured in 2005. We tried to correlate the missing OH reactivity and branched-alkenes reactivity, as shown in Fig 11.

43. L.402-403: consider to change this to: "even with the mixing ratios of the 4 branched alkenes measured in Beijing in 2005, the reactivity…"

Response: Thanks for advice. Accepted and modified as suggested. It is much easier to understand.

44. L.419-420: need a reference for the statement that the mixing ratios of branched alkenes could be lower than 0.1 ppbV. The site in Beijing is only a few hundred meters from major roads and can easily get influenced by vehicle emissions.

Response: Thanks for asking. This was from the later measurements in 2015, which has not been published yet. Also we could achieve similar results if we used the emission ratio method. If we assume the emission ratio of branched-alkenes to chained-alkenes remains constant between 2005 and 2013, we could get the calculated mixing ratios of branched-alkenes, which were lower than 0.1 ppbV.

45. L.425-426 and Table S5: not sure if I understand the "major secondary contributors to modeled reactivity." Why only these species are listed in Table S5? Are they related unmeasured intermediates that are calculated in the model? Need definitions for the acronyms in the model (ALD, DCB, etc.).

Response: Thanks for the questions. These five categories contributed the most among the secondary species modelled in the box model. And thanks for the advice about the explanation for the acronyms, the supplementary files has been revised to include the information.

46. L. 460-462: It's not clear to me how the remaining missing reactivity were allocated into different intermediates. Are these intermediates constrained (remained constant) in the box model run in the second scenario?

Response: Thanks for asking. These intermediates were not constrained in the box model run for the OPE calculation. We just allocate the missing reactivity into these intermediates. As you can see, there could still be missing reactivity between modeled reactivity and measured reactivity. So for the OPE calculation, we are not using the modeled reactivity we've got in 4.3, but just allocate the missing reactivity to different intermediates. However, this will be an important uncertainty for our evaluation.

47. L.466: on average. Also please give absolute values, i.e., increase from XX to YY.

Response: Thanks for the detail. Accepted, and due to the variations of OPE with $NO_2$, we decided to choose 20 ppbV as one example.

48. Section 4.4: again, the OPE needs to be recalculated by including the production rate of organic nitrates in P(NOz). This may change the picture currently shown in Fig.14.

Response: Thanks for advice. Accepted and modified as suggested.

49. L. 497-499: Need to include the modeled OH reactivity in Fig. 11. Without this, it is hard to assess this statement that missing OH reactivity can be reconciled with modeled intermediates that were not measured. Also this statement seems in contrast with the statement in L. 395-399, where the author stated that unmeasured primary VOCs, especially branched alkenes, are responsible for the missing OH reactivity. Please clarify.

Response: Thanks for suggestion and question. For the Fig.11, we have modified as suggested. For the question, I think it was a problem caused by my inappropriate expression. In L.395-399, we are considering the important contribution from unmeasured primary VOCs, especially the branched-alkenes in morning rush hour. However, in L.497-499, we are summarizing the daytime modeling work. The August 17[th] morning was an exception for the evaluating, which left a great missing percentage unsolved.

50. L. 507-510: probably add a sentence stating that efforts to reduce the uncertainty

*of OH reactivity measurements based on the CRM technique to increase the confidence of results as shown in this work.*

Response: Thanks for your suggestions. Accepted and modified.

*51. Fig.5: please give the Grade II of National Ambient Air Quality Standard for ozone and PM2.5 in the caption. Also it seems the two red lines in Fig. 5-a and Fig. 5-b are different, one above 80 ppbV and the other below 80 ppbV. Please clarify.*

Response: Sorry for this mistake. We have modified the figures as you suggested.

*52. Fig.6: the yellow (or brown) lines show the NO fraction (not percentage) in NOx. Please correct it.*

Response: Thanks for this advice. This is due to my misunderstanding.

*53. Fig.11: please plot the modeled OH reactivity, the same as in Fig. 12. Is the gray area along the red line showing the uncertainty of the measurement? If so, please state this in the caption.*

Response: Thanks for suggestions. Figures revised and the explanation added.

*54. Fig.14: references for OPEs in other studies should be given.*

Response: Thanks for advice. Accepted and modified as suggested.

*55. Again, Table S5 and Table S6 are identical.*

Response: Sorry for this careless mistake. Modified as in the new supplement.

*56. P. 3 of the supplement: in Alkenes, ethane should be ethane.*

Response: Thanks for advice and sorry for this mistake.

---

## Author Response (AR2)

| 1                          | Reply to Reviewer's comments                                                                                                                                                                                                                                                                                                                              |
|----------------------------|-----------------------------------------------------------------------------------------------------------------------------------------------------------------------------------------------------------------------------------------------------------------------------------------------------------------------------------------------------------|
| 2                          | General Comments                                                                                                                                                                                                                                                                                                                                          |
| 3
4
5                | Although the figure quality has improaved, the language still needs to be
improved substantially before final publication.                                                                                                                                                                                                                             |
| 6
7
8                | Response: Appreciate your further comments. Figures and sentences have been updated again and wish it would help.                                                                                                                                                                                                                                         |
| 9                          | Detailed Comments:                                                                                                                                                                                                                                                                                                                                        |
| 10                         | 1. Further model detail needs to be provided in the manuscript itself. The reader                                                                                                                                                                                                                                                                         |
| 11                         | shouldn't have to refer to the author's responses for details on the boundary layer                                                                                                                                                                                                                                                                       |
| 12
13                   | height assumed.                                                                                                                                                                                                                                                                                                                                           |
| 14
15
16
17
18 | Response: Thanks for the advice. The sentence was added in the end of section 2.3.1. The boundary layer height was set as constant as 1000 m in the model due to the lack in measurements. This was similar to model setups in Lu et al (2013) and field measurement results in Guo et al (2016).                                                         |
| 19
20
21
22       | 2. The authors suggest that they have run the model for different spin up times and report a difference within 10%, but no detail on the different spin up times tested is provided!                                                                                                                                                                      |
| 23
24
25
26       | Response: Appreciate the suggestion. There was also one more sentence in section 2.3.1 in the latest version of manuscript. Different scenarios with 1 day, 2 days and 3 days spin-up time have been tried while the differences were within 10%.                                                                                                         |
| 27
28
29
30       | 3. I still think it is important to include a time-series of measured, calculated and modelled reactivity somewhere in the manuscript. If there is missing data for the Beijing campaign, just leave gaps in the time-series rather than interpolate. The pie-charts generated from the campaign averages hide a lot of detail - e.g. diurnal             |
| 31
22                   | variation and day to day variability.                                                                                                                                                                                                                                                                                                                     |
| 33
34
35
36
37 | Response: Thanks for the suggestion. The missing data really caused a lot of problems
for the data evaluation. However, we follow the suggestion from the reviewers and
revised Fig 5a and Fig 5b as in the latest version. However, this suggestion was an
important one, especially for the last sentence: the averages hide a lot of details. |
| 38
39                   | 4. Paris is still not included in figure 10b (and has now been removed from fig. 10a, but is still discussed in section 4.1).                                                                                                                                                                                                                             |
| 40                         |                                                                                                                                                                                                                                                                                                                                                           |
| 41
42
43             | Response: Thanks for the suggestion. The sentence including Paris information were deleted in the latest version. Sorry for the careless mistake. The figures remained as in the last version.                                                                                                                                                            |

| 45
46
47                                           | 5. Mention that the NOAA 2005 dataset was from Beijing in the manuscript. Include a comment on the sensitivity of OPE to the species chosen to represent the missing reactivity in section 4.4.                                                                                                                                                                                                                                                                                                                                                                     |
|----------------------------------------------------------|---------------------------------------------------------------------------------------------------------------------------------------------------------------------------------------------------------------------------------------------------------------------------------------------------------------------------------------------------------------------------------------------------------------------------------------------------------------------------------------------------------------------------------------------------------------------|
| 48

57 | Response: Appreciate the suggestion. For the first one, we add "Beijing" in the first
same sentence referring to NOAA measurements in 2005. For the second one, we
rephrased the sentence in section 4.4 as follows:

[revised manuscript text omitted]

$$\mathbf{k}_{OH} = c\mathbf{1} \times k_{Pyr+OH} \times \frac{c\mathbf{3} - c\mathbf{2}}{c\mathbf{1} - c\mathbf{3}} \tag{2-1}$$

Ambient air or synthetic air was introduced at 160 -170 ml min-1 with the total 216 flow 320 - 350 ml min-1(The typical dilution factor was about 2-2.15 depending on 217 the situation). The residence time of air inside the reactor was less than 30 s before 218 they were pumped by the Teflon pump. The typical c1 mixing ratio for pyrrole in 219 Beijing and Heshan measurements were about 60 ppbV and 55 ppbV, while the 220 mixing ratios of OH radicals generated by mercury lamp were about 35 ppbV and 28 221 ppbV. The mixing ratios were quite consistent for either of the campaigns, 222 respectively. Corrections about pseudo-first order kinetics were conducted for both 223

measurements, based on the methods in Sinha et al (2008). The typical correction

factors could be presented as

$$R_{\rm true} = 0.0008 * (R_{\rm mea})^2 + 0.78 * R_{\rm mea} - 0.042$$
(2-2)

227

226

$$R_{\text{true}} = -0.0004 * (R_{\text{mea}})^2 + 0.81 * R_{\text{mea}} - 0.017$$
(2-3)

[revised manuscript text omitted]

| Mainz, summer, 2005 CRM 10.4 - Sinha et al.,2008
German | aris, France winter, 2010 CRM $10\sim130$ $\frac{10\sim54\%}{\text{less than}}$ SO Dolgorouky et al., 2012 | London, summer, 2012 LP-LIF 10-116 20~40% SFOB Whalley et al., 2016 England | ille, France autumn , 2012 $\frac{\text{CRM}}{\text{LP-LIF}} \sim 70$ Reasonable SFO Hansen et al., 2015 agreement | Dunkirk, summer, 2014 CRM 10-130 - Michoud et al., 2015
France | rent studies, the measured reactivity was presented as the averaged results, or ranges of diurnal variations, or the ranges of the whole |           | studies, the calculated reactivity was presented within an uncertainty range, as a percentage reduction or s -1 reduction. | nave been used for the calculated reactivity (following Lou et al., 2010): S = inorganic compounds (CO, NO x , SO 2 etc) plus hydrocarbons | = formaldehyde; O = OVOCs other than formaldehyde; B = BVOCs other than isoprene; | ormation regarding what has been measured or how long it has been measured. |
|------------------------------------------------------------|------------------------------------------------------------------------------------------------------------|-----------------------------------------------------------------------------|--------------------------------------------------------------------------------------------------------------------|-------------------------------------------------------------------|------------------------------------------------------------------------------------------------------------------------------------------|-----------|---------------------------------------------------------------------------------------------------------------------------------------|------------------------------------------------------------------------------------------------------------------------------------------------------------------|-----------------------------------------------------------------------------------|-----------------------------------------------------------------------------|
| Mainz, summer, 2005
German                              | Paris, France winter, 2010                                                                                 | London, summer, 2012
England                                             | Lille, France autumn, 2012                                                                                         | Dunkirk, summer, 2014
France                                   | n different studies, the measured reactiv                                                                                                |           | fferent studies, the calculated reactivity                                                                                            | s that have been used for the calculated                                                                                                                         | (ne); F = formaldehyde; O = OVOCs oth                                             | of information regarding what has been                                      |
|                                                            | MEGAPOLI                                                                                                   | ClearfLo                                                                    |                                                                                                                    |                                                                   | a. For sources from                                                                                                                      | campaign. | b. For sources of d                                                                                                                   | c. Measured specie                                                                                                                                               | (including isopr                                                                  | d. "-" means a lack                                                         |

Table 1 Total OH reactivity measurements in urban areas (continued)

|                            | 1401 | 22 INTAL OIL ICAULI |               | כ אווש וושחוחמון מווא         | uitouituitig atcas                                     |                     |                                            |
|----------------------------|------|---------------------|---------------|-------------------------------|--------------------------------------------------------|---------------------|--------------------------------------------|
| Site                       |      | Year                | method        | $k_{OH(measured)}$ $(s^{-1})$ | kOH (calculated)
(s -1 if it is a value) | Measured
Species | Reference                                  |
| Central
Pennsylvania, U | S    | spring, 2002        | LIF-flow tube | 6.1                           |                                                        |                     | Ren et al., 2005                           |
| Whiteface
Mountain, US  |      | summer,
2002     | LIF-flow tube | 5.6                           | within 10%                                             | ı                   | Ren et al.,
2006b                       |
| Weybourne,
England      |      | spring, 2004        | LIF-flow tube | 4.85                          | 2.95                                                   | SFO                 | Ingham et al.,
2009
Lee et al., 2010 |
| Yufa, China                |      | summer,
2006     | LP-LIF        | 10-30                         | agree well                                             | S                   | Lu et al., 2010;
2013                   |
| Backgarden, China          |      | summer,
2006     | LP-LIF        | 10~120                        | 50% less than                                          | $\mathbf{S}$        | Lou et al., 2010                           |
| El Arenosillo,
Spain    |      | winter, 2008        | CRM           | 6.3~85                        |                                                        | SF                  | Sinha et al.,
2012                      |
|                            |      | spring, 2013        | CRM           | 53                            | 23                                                     | SFOB                | Kumar &
Sinha., 2014                    |
|                            |      |                     |               |                               |                                                        |                     |                                            |

Table2 Total OH reactivity measurements in suburban and surrounding areas

928 Fig 1 Schematic figures of CRM system in Beijing and Heshan observations.

Blue color represents ambient air or synthetic air injection system, purple color

930 represents OH generating system, black color represents the detection system.

931 Pressure is measured by the sensor connected to the glass reaction.